# FUSION IS NOT ENOUGH: SINGLE MODAL ATTACKS ON FUSION MODELS FOR 3D OBJECT DETECTION

**Zhiyuan Cheng**[1]  **Hongjun Choi**[2]  **Shiwei Feng**[1]  **James Liang**[3]  **Guanhong Tao**[1]
**Dongfang Liu**[3]  **Michael Zuzak**[3]  **Xiangyu Zhang**[1]
[1]Purdue University  {cheng443, feng292, taog, xyzhang}@purdue.edu
[2]DGIST  hongjun@dgist.ac.kr
[3]Rochester Institute of Technology  {jcl3689, dongfang.liu, mjzeec}@rit.edu

## ABSTRACT

Multi-sensor fusion (MSF) is widely used in autonomous vehicles (AVs) for perception, particularly for 3D object detection with camera and LiDAR sensors. The purpose of fusion is to capitalize on the advantages of each modality while minimizing its weaknesses. Advanced deep neural network (DNN)-based fusion techniques have demonstrated the exceptional and industry-leading performance. Due to the redundant information in multiple modalities, MSF is also recognized as a general defence strategy against adversarial attacks. In this paper, we attack fusion models from the camera modality that is considered to be of lesser importance in fusion but is more affordable for attackers. We argue that the weakest link of fusion models depends on their most vulnerable modality, and propose an attack framework that targets advanced camera-LiDAR fusion-based 3D object detection models through camera-only adversarial attacks. Our approach employs a two-stage optimization-based strategy that first thoroughly evaluates vulnerable image areas under adversarial attacks, and then applies dedicated attack strategies for different fusion models to generate deployable patches. The evaluations with six advanced camera-LiDAR fusion models and one camera-only model indicate that our attacks successfully compromise all of them. Our approach can either decrease the mean average precision (mAP) of detection performance from 0.824 to 0.353, or degrade the detection score of a target object from 0.728 to 0.156, demonstrating the efficacy of our proposed attack framework. Code is available.

## 1 INTRODUCTION

3D object detection is a critical task in the perception of autonomous vehicles (AVs). In this task, AVs employ camera and/or LiDAR sensors input to predict the location, size, and categories of surrounding objects. Camera-LiDAR fusion models, which combine the high-resolution 2D texture information from camera images with the rich 3D distance information from LiDAR point clouds, have outperformed the detection accuracy of models that rely solely on cameras or LiDAR. (Yang et al., 2022; Liu et al., 2023b; Li et al., 2022b). Additionally, multi-sensor fusion (MSF) techniques are generally recognized as a defensive measure against attacks (Cao et al., 2021; Liang et al., 2022), as the extra modality provides supplementary information to validate detection results. Viewed in this light, a counter-intuitive yet innovative question arises: ❶ *Can we attack fusion models through a single modality, even the less significant one, thereby directly challenging the security assumption of MSF*? Yet, this fundamental question has not been sufficiently answered in the literature.

Previous research has demonstrated successful attacks against camera-LiDAR fusion models by targeting either multiple modalities (Cao et al., 2021; Tu et al., 2021) or the LiDAR modality alone (Hallyburton et al., 2022). However, these approaches are not easy to implement and require additional equipment such as photodiodes, laser diodes (Hallyburton et al., 2022), and industrial-grade 3D printers (Cao et al., 2021; Tu et al., 2021) to manipulate LiDAR data, thus increasing the deployment cost for attackers. Consequently, we explore the possibility of attacking fusion models via the camera modality, as attackers can more easily perturb captured images using affordable adversarial patches. Nevertheless, this attack design presents additional challenges. For example, the camera modality is considered less significant in fusion models for 3D object detection since LiDAR provides abundant 3D information. The performance of both state-of-the-art LiDAR-based models and ablations of fusion models using only LiDAR surpasses their solely camera-based counterparts

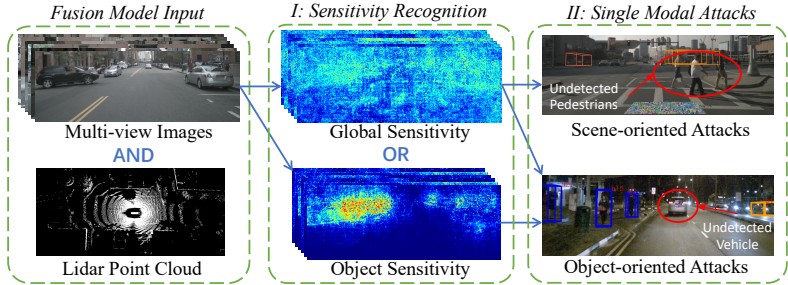

Figure 1: **Single-modal attacks** against camera-LiDAR fusion model using camera-modality.

significantly (Liang et al., 2022; Liu et al., 2023b; Motional, 2023) (see more experimental results in Appendix A). The less significance of camera modality in fusion can limit its impact on detection results. Moreover, different fusion models can exhibit distinct vulnerabilities in the camera modality, necessitating varying attack strategies. The cutting-edge adversarial patch optimization technique against camera-only models (Cheng et al., 2022) has limitations in generating deployable patches viewing the entire scene, as they fail to consider the semantics of the input. Hence, a problem remains open: ❷ *How to design single-modal attack to effectively subvert fusion models?*

In response to ❶ and ❷, we propose a novel attack framework against camera-LiDAR fusion models through the less significant camera modality. We utilize adversarial patches as the attack vector, aiming to cause false negative detection results, and our main focus lies on the early-fusion scheme, including data-level and feature-level fusion strategies. As shown in Figure 1, our attack employs a two-stage approach to generate an optimal adversarial patch for the target fusion model. In the first stage ($2^{nd}$ column), we identify vulnerable regions in the image input using our novel sensitivity distribution recognition algorithm. The algorithm employs an optimizable mask to identify the sensitivity of different image areas under adversarial attacks. Based on the identified vulnerable regions, we then classify the fusion model as either object-sensitive or globally sensitive, enabling tailored attack strategies for each type of model. In the second stage ($3^{rd}$ column), we design two attack strategies for different types of models to maximize attack performance. For globally sensitive models, we devise scene-oriented attacks, wherein adversarial patches can be placed on static background structures (e.g., roads or walls) to compromise the detection of arbitrary nearby objects (see undetected pedestrians in the red circle of Figure 1). For object-sensitive models, we implement object-oriented attacks that can compromise the detection of a target object by attaching the patch to it (see the undetected vehicle in the red circle of Figure 1). Compared to Cheng et al. (2022), the patches generated by our proposed framework offer a significant advantage by being both physically deployable and effective (see comparison in Appendix J). Our contributions are:

- We present single-modal attacks against advanced camera-LiDAR fusion models leveraging only the camera modality, thereby further exposing the security issues of MSF-based AV perception.
- We develop an algorithm for identifying the distribution of vulnerable regions in images, offering a comprehensive assessment of areas susceptible to adversarial attacks.
- We introduce a framework for attacking fusion models with adversarial patches, which is a two-stage approach and involves different attack strategies based on the recognized sensitivity type of the target model. The threat model is detailed in Appendix P.
- We evaluate our attack using six state-of-the-art fusion-based and one camera-only models on Nuscenes (Caesar et al., 2020), a real-world dataset collected from industrial-grade AV sensor arrays. Results show that our attack framework successfully compromises all models. Object-oriented attacks are effective on all models, reducing the detection score of a target object from 0.728 to 0.156 on average. Scene-oriented attacks are effective for two globally sensitive models, decreasing the mean average precision (mAP) of detection performance from 0.824 to 0.353. Experiments in simulation and physical-world also validate the practicality of our attacks in the real world. Demo video is available at `https://youtu.be/xhXtzDezeaM`.

## 2 RELATED WORK

**Camera-LiDAR Fusion.** AVs are typically equipped with multiple surrounding cameras, providing a comprehensive view, and LiDAR sensors are usually mounted centrally on top of the vehicle, enabling a 360-degree scan of the surrounding environment, resulting in a 3D point cloud. Images and point clouds represent distinct modalities, and numerous prior works have investigated methods

to effectively fuse them for improved object detection performance. Specifically, the fusion strategies can be categorized into three types based on the stage of fusion: 1) data-level fusion, which leverages the extracted features from one modality to augment the input of the other modality (Yin et al., 2021; Vora et al., 2020; Wang et al., 2021); 2) decision-level fusion, which conducts independent perception for each modality and subsequently fuses the semantic outputs (BaiduApollo); and 3) feature-level fusion, which combines low-level machine-learned features from each modality to yield unified detection results (Liu et al., 2023b; Liang et al., 2023; Yang et al., 2022; Li et al., 2022b; Bai et al., 2022; Chen et al., 2022b). Feature-level fusion can be further divided into alignment-based and non-alignment-based fusion. Alignment-based fusion entails aligning camera and LiDAR features through dimension projection at the point level (Li et al., 2020; Vora et al., 2020; Chen et al., 2022a), the voxel level (Li et al., 2022b; Jiao et al., 2022), the proposal level (Chen et al., 2017; Ku et al., 2018), or the bird's eye view (Liu et al., 2023b; Liang et al., 2022) before concatenation. For non-alignment-based fusion, cross-attention mechanisms in the transformer architecture are employed for combining different modalities (Yang et al., 2022; Bai et al., 2022). Contemporary fusion models primarily use feature-level fusion for its superior feature extraction capability and performance. Hence, we focus on introducing and analyzing this type of fusion strategy in our method design. It is worth noting that our approach can also be directly applied to data-level fusion, as demonstrated in our evaluation (see Section 5). More discussion of fusion strategies is in Appendix B. Appendix C introduces the general architecture of camera-LiDAR fusion.

**3D Object Detection Attacks.** 3D object detection models (Cheng et al., 2022; Liu et al., 2021a; Cui et al., 2021) can be classified into three categories: camera-based, LiDAR-based, and fusion-based models. Attacks targeting each category have been proposed in the context of AV systems. 1) For camera-based models, adversaries typically employ adversarial textures to manipulate the pixels captured by AV cameras (Zhang et al., 2021; Boloor et al., 2020). This approach is cost-effective and can be easily implemented through printing and pasting an adversarial patch. Recent studies have concentrated on enhancing the stealthiness of the adversarial patterns (Cheng et al., 2022; Duan et al., 2020). 2) In the case of LiDAR-based models, some attackers utilize auxiliary equipment, such as photodiodes and laser diodes, to intercept and relay the laser beams emitted by AV LiDAR systems, thereby generating malicious points in the acquired point cloud to launch the attack (Cao et al., 2019; 2023; Sun et al., 2020). Alternatively, others employ malicious physical objects with engineered shapes to introduce adversarial points in attacks (Tu et al., 2020; Abdelfattah et al., 2021a; Cao et al., 2019). 3) Regarding camera-LiDAR fusion models, multi-modal attacks have been developed that perturb both camera and LiDAR input either separately (Tu et al., 2021; Abdelfattah et al., 2021b) or concurrently (Cao et al., 2021), using the previously mentioned attack vectors. Additionally, single-modal attacks on solely LiDAR input have been conducted in a black-box manner (Hallyburton et al., 2022) to fool fusion models. For camera-oriented single modal attacks, there are prior works investigating the robustness of fusion models when subjected to noisy camera input (Park et al., 2021; Kim & Ghosh, 2019). However, Kim & Ghosh (2019) mainly considered random noise, specifically Gaussian noise, instead of physical-world adversarial attacks. Park et al. (2021) mainly focused on digital-space attacks and exclusively targeted an early model using single-view images. Differently, our study considers physically practical attacks and investigates fusion models utilizing multi-view images and a transformer-based detection head.

## 3 MOTIVATION

Despite the challenges mentioned in Section 1, it is still theoretically possible to conduct camera-only attack against fusion models. The intuition behind is that adversarial effects from the camera modality can propagate through model layers, contaminate the fused features, and ultimately impact the model output (See Appendix D for detailed feasibility analysis). To examine the actual performance of camera-only adversarial attacks on SOTA fusion models, we illustrate an example in Figure 2. A frame is derived from the Nuscenes dataset containing both camera and LiDAR data (the first row). It represents a scenario where the ego-vehicle is navigating a road populated with multiple cars and pedestrians. In be-

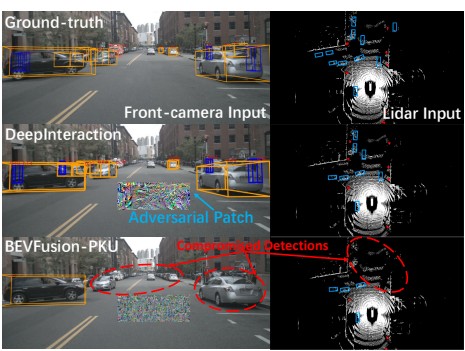

Figure 2: **Motivating example** of adversarial patch attack on images against fusion models.

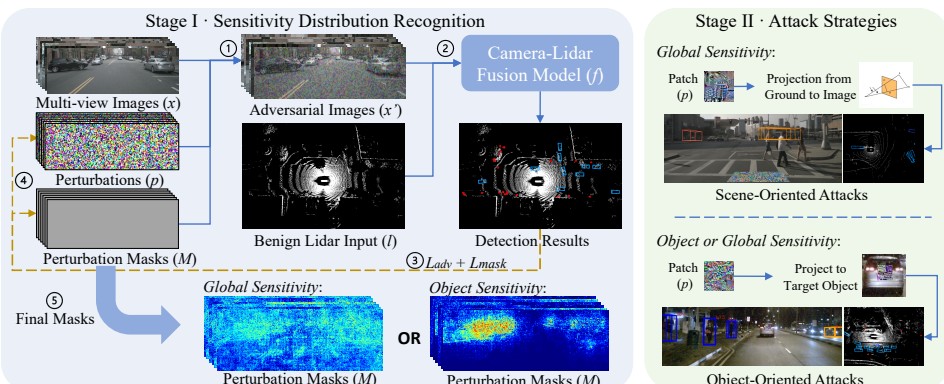

Figure 3: **Framework of single-modal attacks** against camera-LiDAR fusion model with adversarial patches.

nign cases, two cutting-edge fusion models, DeepInteraction (Yang et al., 2022) and BEVFusion-PKU (Liang et al., 2022), can accurately detect objects in the given scene. We then implement a conventional patch attack (Brown et al., 2017) by generating a patch on the road to induce false negative detections. The performance of DeepInteraction is undisturbed by the attack, illustrated in the second row of Figure 2. In contrast, BEVFusion-PKU is successfully disrupted, evidenced by its inability to detect objects proximal to the patch, highlighted by red circles in the third row. This discrepancy in the models' responses confirms that exploiting the camera modality can impact fusion models while highlights that *uniform attack strategies may not be universally effective due to the inherent unique vulnerabilities in different models, such as varying susceptible regions*. Despite the SOTA patch attack can be adapted to optimize the patch region over the scene, the generated patch is not deployable (see Appendix J), limiting its application.

To characterize the susceptible regions, we introduce the concept of "*sensitivity*" as a property of areas in input images. It measures the degree to which specific area of an image impacts adversarial goals. An area with high sensitivity means perturbations there have large influence and can achieve good attack performance. Hence, sensitive regions are more vulnerable to adversarial attacks than other regions. Formally, the sensitivity $S_A$ of an area $A$ is defined as $S_A \propto \max_p\{L_{adv}(x,l) - L_{adv}(x',l)\}$, where $x' = x \odot (1 - A) + p \odot A$. Here, $x$ is the input image, $l$ is the LiDAR point cloud and $x'$ is the adversarial image with perturbations $p$ in region $A$. $L_{adv}$ denotes the adversarial loss defined by adversarial goals. Examining the sensitivity of each area on the image through individual patch optimization is very time consuming and it becomes unaffordable as the granularity of the considered unit area increases. Despite the availability of various interpretation methods for model decisions (e.g., GradCAM (Selvaraju et al., 2017) and ScoreCAM (Wang et al., 2020)), which can generate heatmaps to highlight areas of attention in images, it is essential to distinguish between interpreting model decisions and recognizing sensitivity. For instance, our motivating example presents that the road is a susceptible region for adversarial attacks on BEVFusion-PKU. However, the main focus of object detection should be directed towards the objects themselves rather than the road, as an interpretation method would show (Gildenblat, 2022). Therefore, to recognize the sensitivity distribution on input images efficiently, we propose a novel optimization-based method in the first stage, and design different attack strategies in the second stage to maximize attack performance.

## 4 METHOD

**Overview.** Figure 3 presents the framework of our single-modal adversarial attack on fusion models using an adversarial patch, employing a two-stage approach. Initially, we identify the sensitivity distribution of the subject network, and subsequently, we launch an attack based on the identified sensitivity type. *During the first stage*, to recognize the sensitivity distribution, we define perturbations and perturbation masks with dimensions identical to the multi-view image input. We then compose the adversarial input by applying the patch and mask to images of a scene sampled from the dataset (step ①). After feeding the adversarial input images and corresponding benign LiDAR data to the subject fusion model, we obtain object detection results (step ②). We calculate the adversarial loss based on the detection scores of objects in the input scene (step ③) and utilize back-propagation and gradient descent to update masks and perturbations, aiming to minimize adversarial loss and mask loss (step ④). We repeat this process for thousands of iterations until convergence is

achieved, and then visualize the final mask as a heatmap to determine the sensitivity type (step ⑤). The heatmap's high-brightness regions signify areas more susceptible to adversarial attacks. Based on the distribution of sensitive areas, we classify the fusion model into two types: *global sensitivity* and *object sensitivity*. Global sensitivity refers to the distribution of sensitive areas covering the entire scene, including objects and non-object background. Object sensitivity, on the other hand, indicates that only object areas are sensitive to attacks.

*In the second stage*, we adopt different attack strategies based on the identified sensitivity heatmap type. For global sensitivity, we implement scene-oriented attacks. By placing a patch on the static background (e.g., the road), we deceive the fusion model and compromise the detection of arbitrary objects surrounding the patch. For both object sensitivity and global sensitivity, we can employ object-oriented attacks. In this approach, we attach a patch to a target object, causing failure in detecting it while leaving the detection of other objects unaltered. Since adversarial patches, optimized as 2D images, would be deployed physically during attacks, we employ projections (Cheng et al., 2023) to simulate how the patch would look on the scene image once it is physically deployed (refer to Figure 4), which enhances the physical-world robustness. The two attack strategies differ mainly in their projection functions and the scope of affected objects. Details are discussed later.

**Sensitivity Distribution Recognition.** We leverage the gradients of images with respect to the adversarial loss as an overall indicator to understand the sensitivity of different image areas, since they are closely related to the relative weights assigned to the camera modality. (See detailed analysis in Appendix E.) In a formal setting, the proposed sensitivity distribution recognition algorithm can be articulated as an optimization problem. The primary objective is to concurrently minimize an adversarial loss and a mask loss, which can be mathematically represented as follows:

$$\arg\min_{p,m} L_{adv} + \lambda L_{mask}, \quad \textbf{s.t. } p \in [0,1]^{3 \times h \times w}, m \in R^{1 \times \lfloor h/s \rfloor \times \lfloor w/s \rfloor}, \tag{1}$$

$$\textbf{where } L_{adv} = MSE\left(f_{scores}\left(x',l\right),0\right); \quad L_{mask} = MSE(M,0); \tag{2}$$

$$x' = x \odot (1-M) + p \odot M; \quad M[i,j] = \frac{1}{2} \times \tanh(\gamma \cdot m[\lfloor\frac{i}{s}\rfloor, \lfloor\frac{j}{s}\rfloor]) + \frac{1}{2}. \tag{3}$$

Here, $x$ is the image input, which is normalized and characterized by dimensions $h$ (height) and $w$ (width). The symbols $l$, $p$, $m$, and $\lambda$ represent the LiDAR input, the perturbations on image with dimensions equal to $x$, the initial mask parameters, and the mask loss weight hyperparameter, respectively. The desired sensitivity heatmap corresponds to the perturbation mask $M$. Visualization of variables can be found in Figure 3. Initially, the mask parameters $m \in R^{1 \times \lfloor h/s \rfloor \times \lfloor w/s \rfloor}$ are transformed into the perturbation mask $M \in [0,1]^{1 \times h \times w}$ using Equation 3. We use `tanh()` function to map values in $m$ into the $[0,1]$ range, and its long-tail effect encourages the mask $M$ values to gravitate towards either 0 or 1. The hyperparameters $\gamma$ and $s$ modulate the convergence speed and heatmap granularity, respectively. Subsequently, the perturbation mask $M$ is utilized to apply the perturbation $p$ to the input image $x$, resulting in the adversarial image $x'$. $\odot$ denotes element-wise multiplication. Adversarial image $x'$ and benign LiDAR data $l$ are then feed to the fusion model $f_{scores}$. Since our attack goals are inducing false negative detection results, one objective of our optimization is to minimize the detected object scores. Hence, we use the mean square error (MSE) between the scores and zero as the adversarial loss $L_{adv}$. In this context, the output of $f_{scores}$ consists of the detected object scores (confidence). The optimization's secondary objective is to minimize the perturbation mask values, achieved by incorporating a mask loss $L_{mask}$.

The optimization of these two losses is a dual process. Minimizing the adversarial loss (i.e., maximizing attack performance) necessitates a higher magnitude of perturbations on the input. Conversely, minimizing the mask loss indicates a lower magnitude of perturbations. As a result, the dual optimization process converges on applying higher magnitude perturbations on sensitive areas (to improve attack performance) and lower magnitudes for insensitive parts (to minimize mask loss). Hence, the mask $M$ serves as a good representation of the sensitivity distribution, and visualizing $M$ allows for the attainment of the sensitivity heatmap. Then we can further classify the fusion model into object sensitivity or global sensitivity by comparing the expectation of the average intensity of object areas with non-object background in each scene as follows:

$$T(f) = \begin{cases} Object, \mathbf{E}_x \left[\frac{\sum(M^x \odot A_o^x)}{\sum A_o^x}\right] > \beta \mathbf{E}_x \left[\frac{\sum(M^x \odot (1-A_o^x))}{\sum(1-A_o^x)}\right] \\ Global, \qquad\qquad\qquad otherwise \end{cases}. \tag{4}$$

Here, $T(f)$ represents the sensitivity type of fusion model $f$, and $A_o^x$ is a mask with values 1 for object areas and 0 for non-object areas in scene image $x$. The mask $A_o^x$ is generated by considering the

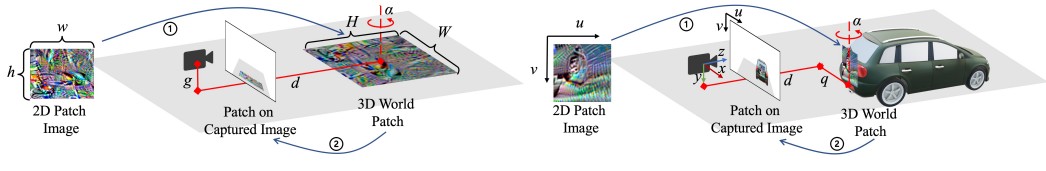

(a) Scene-oriented Attacks.      (b) Object-oriented Attacks.

Figure 4: **Projections** in different attack strategies.

pixels covered by bounding boxes of detected objects in benign cases. $M^x$ refers to the recognized sensitivity heatmap of $x$. $\beta$ is the classification threshold and set to 3 in our experiments.

**Attack Strategies.** Two attack strategies, namely scene-oriented attacks and object-oriented attacks, are introduced based on the fusion model's sensitivity type. Both strategies employ optimization-based patch generation methods. Back-propagation and gradient descent are utilized iteratively to solve the optimization problem. Formally, the problem is defined as:

$$\arg\min_{p} \ \mathbf{E}_{(x,l)\sim D}\left[MSE(f_s(x',l),0)\right], \ \textbf{s.t.} \ p \in [0,1]^{3\times h\times w}, M \in \{0,1\}^{1\times h\times w}, \quad (5)$$

$$\textbf{where} \ \ x' = x \odot (1 - M_x) + p_x \odot M_x; \ \ M_x = proj_x(M); \ \ p_x = proj_x(p). \quad (6)$$

Here, scene images $x$ and LiDAR data $l$ are randomly sampled from the training set $D$. The mask $M$ represents a patch area for cropping the patch image, with values equal to 1 inside a predefined patch area and 0 elsewhere. Unlike Equation 1, $M$ contains discrete values and is not optimizable. $proj_x()$ signifies the projection of the original patch image (see the "2D patch image" in Figure 4a) onto a specific area of the scene image $x$ (see the "captured image" in Figure 4a) to simulate how the patch would look once it's physically deployed, which minimizes the disparity between digital space and the physical world. The target region is contingent upon the attack strategy. Similarly, the output of the fusion model $f_s$ consists of detected object scores, which vary in scope depending on specific attack strategies. We minimize the MSE between detected object score(s) and zero to achieve false negative detection results, and we leverage the Expectation of Transformation (EoT) (Athalye et al., 2018) across all training samples and color jitters (i.e., brightness, contrast and saturation changes) in the patch to enhance the physical robustness and generality of our attack. The adversarial pattern can be concealed within natural textures (e.g., dirt or rust), utilizing existing camouflage techniques (Duan et al., 2020) to remain stealthy and persistent, avoiding removal.

Specifically, for *scene-oriented attacks*, the goal is to compromise the detection of arbitrary objects near an adversarial patch attached to static structures (e.g., the road) of a target scene. In this scenario, the training set $D$ is composed of the target scene in which the ego-vehicle is stationary (e.g., stop at an intersection or a parking lot). The categories and locations of objects surrounding the ego-vehicle in the scene can change dynamically. The output of the fusion model $f_s$ during optimization is the detection score of *all* detected objects in the target scene. To simulate the appearance of the patch on the scene image once deployed, $proj_x$ first projects pixels of the patch image and mask into 3D space on the road (step ① in Figure 4a). Then the function maps them back onto the scene image (step ②). The patch's 3D position is predefined with distance $d$ and viewing angle $\alpha$ by the attacker. The victim vehicle's camera height $g$ above ground can be known from the dataset, which ensures by definition that the patch is on the road. This process can be expressed formally with Equation 7 and 9, where $(u^p, v^p)$ is a pixel's coordinates on the patch image $p$, $(x^p, y^p, z^p)$ the 3D coordinates of the pixel on the ground in the camera's coordinate system, $(u^s, v^s)$ the corresponding pixel on the scene image, $K$ the camera intrinsic parameters. Other variables are defined in Figure 4.

$$\begin{bmatrix} x^p \\ y^p \\ z^p \\ 1 \end{bmatrix} = \begin{bmatrix} \cos\alpha & 0 & -\sin\alpha & q \\ 0 & 1 & 0 & 0 \\ \sin\alpha & 0 & \cos\alpha & d \\ 0 & 0 & 0 & 1 \end{bmatrix} \cdot \begin{bmatrix} W/w & 0 & -W/2 \\ 0 & 0 & g \\ 0 & -H/h & H/2 \\ 0 & 0 & 1 \end{bmatrix} \cdot \begin{bmatrix} u^p \\ v^p \\ 1 \end{bmatrix}, \quad (7)$$

$$\begin{bmatrix} x^p \\ y^p \\ z^p \\ 1 \end{bmatrix} = \begin{bmatrix} \cos\alpha & 0 & -\sin\alpha & q \\ 0 & 1 & 0 & 0 \\ \sin\alpha & 0 & \cos\alpha & d \\ 0 & 0 & 0 & 1 \end{bmatrix} \cdot \begin{bmatrix} W/w & 0 & -W/2 \\ 0 & H/h & -H/2 \\ 0 & 0 & 0 \\ 0 & 0 & 1 \end{bmatrix} \cdot \begin{bmatrix} u^p \\ v^p \\ 1 \end{bmatrix}, \quad (8)$$

$$[u^s \ v^s \ 1]^\top = 1/z^p \cdot K \cdot [x^p \ y^p \ z^p \ 1]^\top. \quad (9)$$

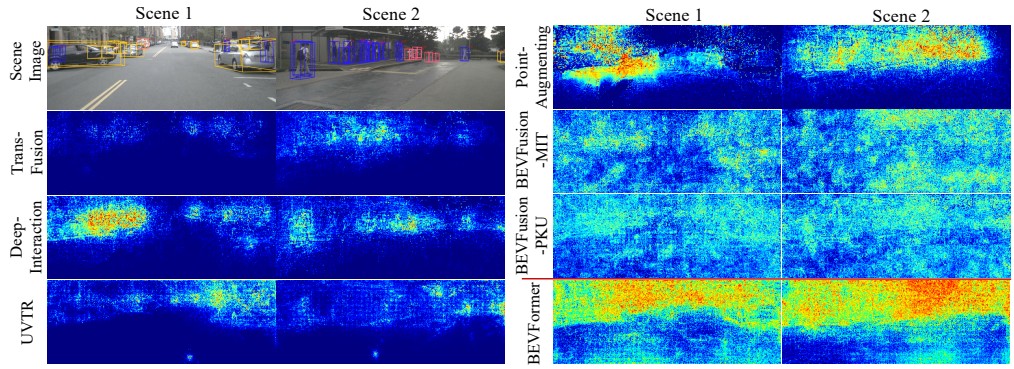

Figure 5: **Sensitivity heatmaps** of six camera-LiDAR fusion models and a camera-only model on two scenes.

For *object-oriented attacks*, the goal is to compromise the detection of the target object with an attached adversarial patch while keeping other objects unaffected. In this case, the training set $D$ is composed of frames in which the target object appears. For example, the ego-vehicle may follow a target vehicle with the background changes dynamically in the scene. The output of the fusion model $f_s$ during optimization is the detection score of the target object exclusively. The function $proj_x$ projects the patch image and mask onto the target object in the scene image using Equation 8 and 9, corresponding to step ① and ② in Figure 4b respectively. Unlike the scene-oriented attack, in which the location of the patch is defined by attackers using longitudinal distance $d$, lateral distances $q$ and viewing angle $\alpha$, in object-oriented attacks, these projection parameters change dynamically depending on the position of the target object in training data. Hence, we innovatively extract them from the estimated 3D bounding box of the target object before projecting the patch.

## 5 EVALUATION

**Model selection** In our evaluation, we use six state-of-the-art camera-LiDAR fusion-based 3D object detection models that are published recently. These models include Transfusion (Bai et al., 2022), DeepInteraction (Yang et al., 2022), UVTR (Li et al., 2022b), PointAugmenting (Wang et al., 2021), BEVFusion-MIT (Liu et al., 2023b) and BEVFusion-PKU (Liang et al., 2022). These models cover data-level and feature-level fusion strategies and contain a diverse range of feature-level fusion approaches, including alignment-based fusion, non-alignment-based fusion, and various detection head designs. Additionally, we use a camera-only model called BEVFormer (Li et al., 2022c) as comparison[1]. Detailed selection criteria can be found in Appendix F.

**Scene selection.** Our evaluation scenes are selected from the Nuscenes dataset (Caesar et al., 2020). This dataset contains real-world multi-view images and point cloud data collected from industrial-grade sensor array, and they are derived from hundreds of driving clips. The selected scenes for testing in our evaluation contains 375 data frames, encompass diverse road types, surrounding objects and time-of-day situations. Additionally, we conduct experiments in simulation and in the physical world. By leveraging this rich dataset along with simulation and physical-world experiments, our evaluation framework benefits from an accurate representation of real-world driving scenarios.

### 5.1 SENSITIVITY DISTRIBUTION RECOGNITION

This section reports on the evaluation of our sensitivity distribution recognition method. We present the qualitative results of the sensitivity heatmap generated by our method and validate the property of the heatmap in Appendix G. We utilize Equation 1 to generate the sensitivity heatmap for the six fusion models, using two different data frames, each with varying proportions of vehicles and pedestrians. Detailed experimental setups can be found in Appendix H. Figure 5 depicts the generated sensitivity heatmaps. The first two images are the scene images captured by the front camera of the ego vehicle while the subsequent rows exhibit the sensitivity distributions of the corresponding scene image using different models. The brightness or warmth of colors in the heatmap corresponds to the sensitivity of a particular region to adversarial attacks. Higher brightness areas signify higher susceptibility to attacks, while lower brightness denotes more robustness. Observe that the sensitive regions for the initial four models, namely Transfusion (Bai et al., 2022), DeepInteraction (Yang et al., 2022), UVTR (Li et al., 2022b) and PointAugmenting (Wang et al., 2021), primarily lie on

---

[1]The code is available at: https://github.com/Bob-cheng/CL-FusionAttack

Table 1: Attack performance of the **scene-oriented adversarial patch attack** against 3D object detection.

| Models | | mAP | CR | TK | BS | TR | BR | PD | BI |
|---|---|---|---|---|---|---|---|---|---|
| BF-PKU | Ben. | 0.824 | 0.453 | 0.448 | 1.000 | 0.991 | 0.898 | 0.990 | 0.989 |
| | Adv. | 0.353 | 0.136 | 0.116 | 0.524 | 0.239 | 0.611 | 0.242 | 0.604 |
| | Diff | 57.2% | 70.0% | 74.1% | 47.6% | 75.9% | 32.0% | 75.6% | 38.9% |
| BF-MIT | Ben. | 0.886 | 0.538 | 0.939 | 0.858 | 0.992 | 0.895 | 0.989 | 0.990 |
| | Adv. | 0.553 | 0.279 | 0.652 | 0.720 | 0.488 | 0.623 | 0.337 | 0.772 |
| | Diff | 37.6% | 48.1% | 30.6% | 16.1% | 50.8% | 30.4% | 65.9% | 22.0% |
| TF | Ben. | 0.758 | 0.493 | 0.451 | 0.700 | 0.991 | 0.692 | 0.989 | 0.990 |
| | Adv. | 0.759 | 0.494 | 0.452 | 0.706 | 0.992 | 0.693 | 0.989 | 0.989 |
| | Diff | 0.1% | 0.2% | 0.2% | 0.9% | 0.1% | 0.1% | 0.0% | 0.1% |
| DI | Ben. | 0.807 | 0.459 | 0.522 | 0.947 | 0.990 | 0.750 | 0.989 | 0.989 |
| | Adv. | 0.808 | 0.460 | 0.529 | 0.947 | 0.990 | 0.751 | 0.989 | 0.989 |
| | Diff | 0.1% | 0.2% | 1.3% | 0.0% | 0.0% | 0.1% | 0.0% | 0.0% |
| UVTR | Ben. | 0.850 | 0.557 | 0.989 | 0.754 | 0.990 | 0.736 | 0.982 | 0.989 |
| | Adv. | 0.862 | 0.558 | 0.989 | 0.786 | 0.990 | 0.741 | 0.982 | 0.989 |
| | Diff | 1.4% | 0.2% | 0.0% | 4.8% | 0.0% | 2.6% | 0.7% | 0.0% |
| PointAug | Ben. | 0.724 | 0.471 | 0.466 | 0.683 | 0.992 | 0.714 | 0.984 | 0.981 |
| | Adv. | 0.716 | 0.467 | 0.468 | 0.679 | 0.988 | 0.705 | 0.984 | 0.981 |
| | Diff | 1.1% | 0.8% | 0.4% | 0.6% | 0.4% | 1.3% | 0.0% | 0.0% |
| BFM | Ben. | 0.519 | 0.417 | 0.811 | 0.280 | 0.247 | 0.712 | 0.650 | 0.518 |
| | Adv. | 0.514 | 0.432 | 0.799 | 0.284 | 0.247 | 0.711 | 0.605 | 0.518 |
| | Diff | 1.1% | 3.6% | 1.5% | 1.4% | 0.0% | 0.1% | 6.9% | 0.0% |

\* CR: Car, TK: Truck, BS: Bus, TR: Trailer, BR: Barrier, PD: Pedestrian, BI: Bicycle.

Table 2: Attack performance of the **object-oriented adversarial patch attack**.

| Models | Targeted object | | | Other objects | | |
|---|---|---|---|---|---|---|
| | Ben. Score | Adv. Score | Diff. | Ben. mAP | Adv. mAP | Diff. |
| TF | 0.655 | 0.070 | 89.24% | 0.921 | 0.923 | 0.30% |
| DI | 0.658 | 0.110 | 83.32% | 0.964 | 0.965 | 0.13% |
| UVTR | 0.894 | 0.189 | 78.83% | 0.963 | 0.963 | 0.00% |
| PointAug | 0.734 | 0.177 | 75.89% | 0.954 | 0.955 | 0.10% |
| BF-MIT | 0.714 | 0.219 | 69.37% | 0.965 | 0.968 | 0.34% |
| BF-PKU | 0.712 | 0.168 | 76.38% | 0.956 | 0.958 | 0.13% |
| Average | 0.728 | 0.156 | 78.63% | 0.954 | 0.955 | 0.17% |
| BFM | 0.955 | 0.095 | 90.02% | 0.578 | 0.571 | 1.08% |

\* TF: TransFusion (Bai et al., 2022), DI: DeepInteraction (Yang et al., 2022), UVTR (Li et al., 2022b), PointAug: PointAugmenting (Wang et al., 2021), BF-PKU: BEVFusion-PKU (Liang et al., 2022), BF-MIT: BEVFusion-MIT (Liu et al., 2023b), BFM: BEVFormer (Li et al., 2022c)

Table 3: **Physical-world attack** performance.

| Pedestrian ID | Original | Benign | Adversarial | Difference |
|---|---|---|---|---|
| 1 | 0.685 | 0.693 | 0.194 | 72.01% |
| 2 | 0.674 | 0.642 | 0.219 | 65.89% |
| 3 | 0.659 | 0.681 | 0.237 | 65.20% |
| Average | 0.673 | 0.672 | 0.217 | 67.76% |

areas of objects like vehicles and pedestrians. This suggests that attacks on objects could prove to be more effective, whereas non-object areas such as the road and walls are more resistant. The following two models (i.e., BEVFusion-MIT (Liu et al., 2023b) and BEVFusion-PKU (Liang et al., 2022)) demonstrate high sensitivities throughout the entire scene, irrespective of objects or background regions. This indicates their vulnerability at a global level. Our technique also works on camera-only models. As shown in the last row, the camera-only model (i.e., BEVFormer (Li et al., 2022c)) demonstrates higher sensitivity in the object area and it is also classified as object sensitivity according to Equation 4. Since different sensitivity types demonstrate distinct vulnerability patterns, we discuss the reason behind in our defense discussion (Appendix N).

## 5.2 SCENE-ORIENTED ATTACKS

Scene-oriented attacks are primarily aimed at fusion models with global sensitivity. Such models are vulnerable to adversarial patches placed on non-object background structures (e.g., the road). Our attack is universal, which can affect the detection of arbitrary dynamic objects in a given scene, even those that were not initially present during patch generation (training). Therefore, our attack is more practical in real-world scenarios as attackers can effortlessly paste generated patches onto the ground, rendering victim vehicles in close proximity blind. This could pose a great risk to pedestrians and surrounding vehicles. Detailed experimental setups can be found in Appendix H.

Table 1 presents the quantitative results of our evaluation and qualitative examples can be found in Appendix I. In Table 1, the first column shows various models, the third column presents the mAP of object detection results in the test set, and the subsequent columns denote the average precision (AP) of different objects categories. We report the subject model's benign performance (no patch), adversarial performance (patch applied) and their difference in percentage (attack performance) for each model. Our findings indicate that the detection accuracy of the two globally sensitive models (i.e., BEVFusion-PKU and BEVFusion-MIT) has considerably decreased, for all object categories. The mAP decreased more than 35%. However, the other five models with object sensitivity remain unaffected. These results align with our conclusion in Section 5.1 and further reveal the vulnerability of globally sensitive models to more influential scene-oriented attacks. Additionally, our experiment confirms the robustness of object-sensitive models under attacks in non-object background areas. In comparison, the camera-based model (i.e., BEVFormer) demonstrates worse benign performance than all fusion-based models, but it is also robust to scene-oriented attacks due to its object-sensitive nature. Demo video is available at https://youtu.be/xhXtzDezeaM.

## 5.3 OBJECT-ORIENTED ATTACKS

Object-oriented attacks target object-sensitive models that are more robust to attacks in non-object background areas. The influence of this attack is more localized, as opposed to the scene-oriented

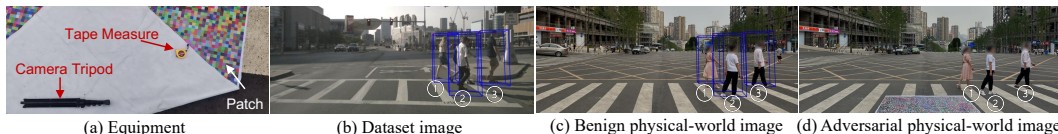

(a) Equipment     (b) Dataset image     (c) Benign physical-world image     (d) Adversarial physical-world image

Figure 6: Attacks in the physical world.

attacks. It concentrates the impact on a specific target object, leaving the detection of other objects unaltered. This approach offers a higher degree of customization for attackers, enabling them to manipulate the impact at the object level rather than the entire scene. Detailed experimental setups can be found in Appendix H. Our evaluation results are presented in Table 2 and qualitative examples are in Appendix I. As shown, the first column represents various fusion models and a camera-only model for comparison, the second to fourth columns display the average detection score of the target object, and the fifth to seventh columns indicate the mAP of other objects (including car, bus, pedestrian and motorcycle). The results demonstrate a substantial decrease in the target object's detection scores for fusion models, from 0.728 to 0.156 on average, thus validating the efficacy of our object-oriented adversarial attacks across all models regardless of the fusion strategies. Furthermore, the detection results of other objects in the scene remain virtually unaffected, as evidenced by the negligible change in mAP. This phenomenon also holds for the camera-only model, and it shows worse benign mAP and more performance degradation under attack. Videos of the attack can be found at https://youtu.be/xhXtzDezeaM.

## 5.4 PRACTICALITY

To assess the practicality of our single-modal attacks on fusion models, we conducted experiments in both simulated and physical-world environments. Attacks in simulation can be found in Appendix K. We assess the feasibility of our attack in a real-world setting by replacing the front-view images of 30 data frames in the dataset with our custom images taken in the physical world, leaving other views and LiDAR data unchanged. To ensure the compatibility of the dataset's LiDAR data with our custom images, we maintain the 3D geometry of our physical scenario consistent with the original dataset images. Figure 6b and Figure 6c illustrate an original image and a custom scenario in our experiment respectively. Note that both images maintain similar 3D geometry, with pedestrians crossing the road located at similar positions in both cases. Detailed setups are in Appendix H. Figure 6a exhibits the experimental equipment, while Table 3 details the attack performance. The pedestrian ID, corresponding to the pedestrians in Figure 6, is denoted in the first column of Table 3, with the subsequent columns reporting the average detection scores in original dataset images (Figure 6b), our benign physical-world images (Figure 6c), and the adversarial physical-world images (Figure 6d). The last column denotes the difference between benign and adversarial physical-world performance. The comparative detection scores in our benign physical-world images and the original dataset images validate the consistency between the original LiDAR data and our custom images, thereby substantiating the efficacy of our image replacement method. Furthermore, the deployment of the adversarial patch results in a significant reduction in the pedestrian detection scores, emphasizing the practicality and effectiveness of our attack strategy in the physical world. We discuss the implications on AV security in Appendix Q.

**Ablation Studies and Defense Discussions.** We conducted ablation studies about the attack performance of various distance and viewing angles of the adversarial patch (Appendix L), and various granularity of the sensitivity heatmap (Appendix M). We discussed both architectural-level defense and DNN-level defense strategies in Appendix N, and the limitations are discussed in Appendix O.

## 6 CONCLUSION

We leverage the affordable adversarial patch to attack the less significant camera modality in 3D object detection. The proposed optimization-based two-stage attack framework can provide a comprehensive assessment of image areas susceptible to adversarial attacks through a sensitivity heatmap, and can successfully attack six state-of-the-art camera-LiDAR fusion-based and one camera-only models on a real-world dataset with customized attack strategies. Results show that the adversarial patch generated by our attack can effectively decrease the mAP of detection performance from 0.824 to 0.353 or reduce the detection score of a target object from 0.728 to 0.156 on average.

## 7 ETHICS STATEMENT

Most of our experiments are conducted in the digital space or in a simulated environment. Our physical-world study involving human subjects underwent thorough scrutiny and approval by an institutional IRB. Notably, we conducted physical experiments in a controlled environment on a closed road, utilizing a camera and tripod to capture scenes instead of employing real cars, as elucidated in Appendix H. This deliberate choice minimizes potential threats to the safety of participants. Stringent protocols were implemented, including participants not facing the camera, wearing masks during experiments, and blurring their faces in the photos. No identifiable information from the volunteers is retained by the researchers.

## 8 ACKNOWLEDGEMENTS

This research was supported, in part by IARPA TrojAI W911NF-19-S-0012, NSF 2242243, 1901242 and 1910300, ONR N000141712045, N00014-1410468 and N000141712947, National Research Foundation of Korea(NRF) grant funded by the Korean government (MSIT) RS-2023-00209836.

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

# Appendix

## Fusion is not Enough: Single Modal Attacks on Fusion Models for 3D Object Detection

This document provides more details about our work and additional experimental settings and result. We organize the content of our appendix as follows:

- Section A: Evaluation of varying significance of different modalities in fusion.
- Section B: Discussion of other fusion strategies.
- Section C: General architecture of Camera-LiDAR Fusion.
- Section D: Feasibility analysis of single-modal attacks against fusion models.
- Section E: Gradient-based analysis of sensitivity distribution recognition.
- Section F: Model selection criteria.
- Section G: Property validation of sensitivity heatmap.
- Section H: Detailed experimental setups.
- Section I: Qualitative results of attacks.
- Section J: Comparison with patch attack against camera-only models.
- Section K: Additional practicality validation.
- Section L: Varying distance and viewing angles.
- Section M: Varying granularity of sensitivity heatmap.
- Section N: Defense discussion.
- Section O: Limitations.
- Section P: Threat model for our attacks.
- Section Q: Discussion on autonomous vehicle security.

## A    VARYING SIGNIFICANCE OF DIFFERENT MODALITIES IN FUSION

LiDAR and camera sensors contribute uniquely to 3D object detection, offering depth and texture information, respectively. These modalities are integrated into fusion models, where their relative importance may depend on their individual contributions to the final output. *LiDAR is often considered more critical for 3D object detection due to its rich 3D information.* Evidence supporting this can be found in Table 4, which presents the reported performance of five state-of-the-art fusion networks, each trained with different modalities. The LiDAR-based models have performance close-to their fusion-based counterparts, outperforming the camera-based models significantly. Further supporting this point, Figure 7 demonstrates that the performance of fusion-based models is considerably more affected by a reduction of information in LiDAR input than in camera input, supporting the greater contribution of the LiDAR modality in detection performance. In this controlled experiment, we reduce the input information by introducing random noise. For the camera, we randomize values of uniformly sampled pixels in input images, and for the LiDAR, we randomize the

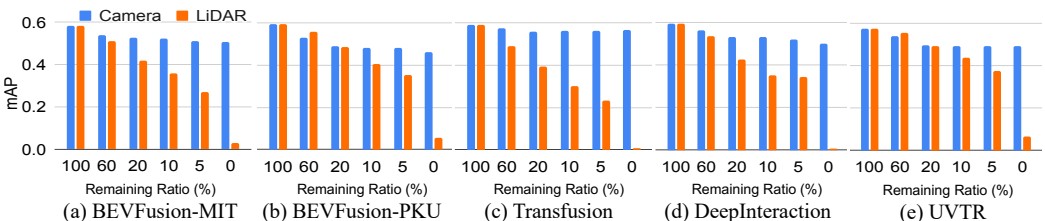

Figure 7: Performance degradation of fusion-based models when camera and LiDAR information are reduced respectively.

positions of uniformly sampled points in the input point cloud. The detection performance is evaluated on the Nuscenes dataset (mini) (Caesar et al., 2020). Only one modality is altered at a time. The results of these investigations clearly demonstrate that the camera modality's contribution to the fusion process is of lesser significance, which implies that mounting an attack via a less influential modality, such as the camera, may prove to be a challenging endeavor.

Table 4: Performance comparison of single modal models and fusion models. Metrics: mAP.

|  | BEVFusion-MIT (Liu et al., 2023b) | BEVFusion-PKU (Liang et al., 2022) | TransFusion (Bai et al., 2022) | DeepInteraction (Yang et al., 2022) | UVTR (Li et al., 2022b) |
|---|---|---|---|---|---|
| C | 0.333 | 0.227 | - | - | 0.362 |
| L | 0.576 | 0.649 | 0.655 | 0.692 | 0.609 |
| CL | 0.664 | 0.679 | 0.689 | 0.699 | 0.654 |

\* C: Camera, L: LiDAR, CL: Camera-LiDAR Fusion,

## B    DISCUSSION OF OTHER FUSION STRATEGIES

For the data-level fusion strategy, some work (Yin et al., 2021) projects the pixels on images into 3D space to create virtual points and integrate with LiDAR points, then uses point cloud-based models on the integrated points for 3D object detection. Perturbations on the camera modality would directly affect the projected virtual points (Cheng et al., 2022). Since point cloud-based models are vulnerable to point-manipulation attacks (Cheng et al., 2022; Cao et al., 2019; Sun et al., 2020; Jin et al., 2023; Liu et al., 2023a; Hau et al., 2021), it is easy to compromise this kind of fusion models through the camera modality as the virtual points projected from images are the weakest link for such models. Other works (Wang et al., 2021; Vora et al., 2020; Qi et al., 2018; Li et al., 2023) use the extracted image features to augment the point data. For example, Wang et al. (2021) utilizes image features extracted by CNN to enhance LiDAR points. In Qi et al. (2018) and Li et al. (2023), authors employ CNN-based 2D object detection or semantic segmentation to identify object regions in the image input, then use the extruded 3D viewing frustum to extract corresponding points from the LiDAR point cloud for bounding box regression. Adversarial attacks on images can deceive the object detection or semantic segmentation result as in Huang et al. (2020); Xie et al. (2017), leading to the failure of the extruded 3D frustum in capturing the corresponding LiDAR points of each object. This would subsequently compromise the 3D bounding box regression. Our evaluation with Wang et al. (2021) has shown that data-level fusion is also vulnerable to adversarial attacks on the camera modality. Hence, our attack techniques are effective for both data-level fusion and feature-level fusion strategies.

The decision-level fusion is less adopted in advanced fusion models. It fuses the prediction results from two separate single-modal models in a later stage. Our camera-oriented attacks will only affect the result of the camera-based model, and the fusion stage can be completed in tracking (BaiduApollo). The attack performance heavily depends on the relative weights assigned to the two modalities. This dependency is a common limitation for single-modal attacks. Although prior LiDAR-only attacks (Hallyburton et al., 2022) have succeeded on Baidu Apollo (BaiduApollo), it can be attributed to the current higher weighting of LiDAR results in Apollo. Some anomaly detection algorithms like Quinonez et al. (2021) may leverage the inconsistency between two modalities' output to alert attacks, but which one to trust and operate with remains uncertain. Developing robust fusion models to defend all kinds of single-modal attacks is an open research question and we leave it as our future work.

## C    GENERAL ARCHITECTURE OF CAMERA-LiDAR FUSION

Figure 8 illustrates the general architecture of cutting-edge camera-LiDAR fusion models using feature-level fusion. From left to right, multi-view images obtained from cameras and the point cloud data from LiDAR sensors are initially processed independently by neural networks to extract image and LiDAR features. The networks responsible for feature extraction are commonly referred to as "*backbones*", which encompass ResNet50 (He et al., 2016), ResNet101 (He et al., 2016), SwinTransformer (Liu et al., 2021b) for images, and SECOND (Yan et al., 2018), PointNet (Qi

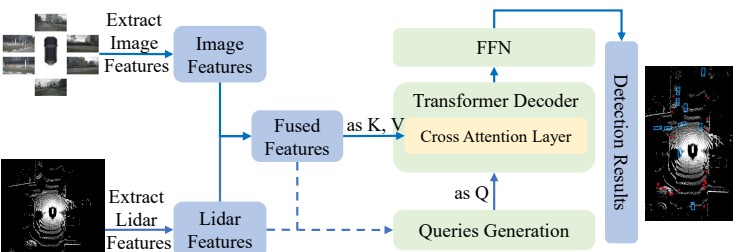

Figure 8: A **general architecture** of concurrent camera-LiDAR fusion models for 3D object detection. FFN denotes Feed-Forward Network.

et al., 2017), VoxelNet (Zhou & Tuzel, 2018) for point cloud data, among others. Subsequently, the extracted features from each modality are fused together employing either alignment-based or non-alignment-based designs, which vary from model to model. After fusion, a detection head is employed to generate the final predictions. The 3D object detection head generally adopts a transformer decoder-based architecture in cutting-edge fusion models since the efficacy of transformers in object detection has been substantiated by DETR (Carion et al., 2020). In the transformer-based detection head, the input consists of three sequences of feature vectors named queries (Q), keys (K) and values (V). Each input query vector corresponds to an output vector, representing detection results for an object, including bounding box, object category, and detection score. Input keys and values, derived from fused features, provide scene-specific semantic information.

The initial queries generation in various models exhibits distinct design characteristics. For instance, UVTR (Li et al., 2022b) uses learnable parameters as queries, DeepInteraction (Yang et al., 2022) and TransFusion (Bai et al., 2022) employ LiDAR features sampled using fused features, while BEVFusion-MIT (Liu et al., 2023b) and BEVFusion-PKU (Liang et al., 2022) utilize bird's-eye-view fused features. The decoder output is subsequently processed by a Feed-Forward Network (FFN) for final regression and classification results. Our attack is general to both alignment-based and non-alignment-based fusion approaches regardless of the design of detection head.

## D  FEASIBILITY ANALYSIS

We analyze the feasibility of single-modal attacks on fusion models in this section. Since modern camera-LiDAR fusion models are all based on DNNs, we start with a simplified DNN-based fusion model. As shown in Figure 9, we use $A^\top = [a_1, a_2]$ to represent the extracted image feature vector and use $B^\top = [b_1, b_2, b_3]$ to denote the LiDAR feature vector. Suppose these features are concatenated to a unified vector during fusion and used to calculate the next layer of features $C^\top = [c_1, c_2, c_3]$ with weight parameters $W \in R^{3 \times 5}$. Hence we have:

$$C^\top = W \cdot \text{vstack}(A, B), \tag{10}$$

where `vstack()` is a concatenation operation. Specifically, the elements of $C$ are calculated as follows:

$$c_i = \sum_{j=1}^{2} w_{i,j} \cdot a_j + \sum_{j=1}^{3} w_{i,2+j} \cdot b_j \ \ (i = 1, 2, 3). \tag{11}$$

Now, suppose adversarial perturbations are applied to an area of the input image and some image features (i.e., $a_1$) are affected while LiDAR features remain benign. Let $A'^\top = [a_1 + \Delta_1, a_2]$ be the adversarial image features. Then the fused features are $C'^\top = [c'_1, c'_2, c'_3]$ calculated as follows:

$$
\begin{aligned}
c'_i &= \sum_{j=1}^{2} w_{i,j} \cdot a_j + \sum_{j=1}^{3} w_{i,2+j} \cdot b_j + w_{i,1}\Delta_1 \\
&= c_i + w_{i,1}\Delta_1 \ \ (i = 1, 2, 3).
\end{aligned} \tag{12}
$$

As we show, *every fused feature is tainted by the adversarial features and the effect will finally propagate to the 3D object detection results, making single-modal attacks on prediction results*

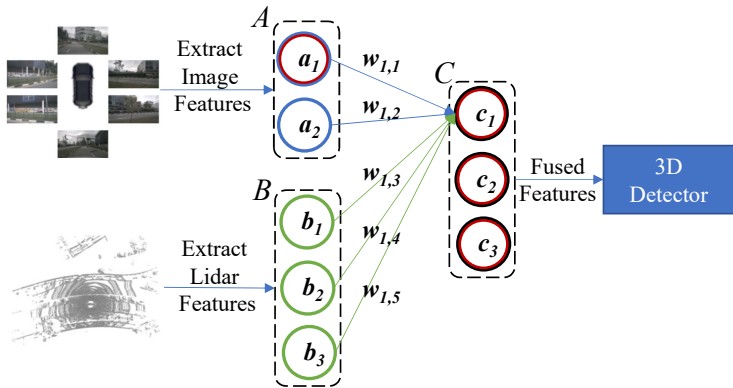

Figure 9: **Simplified illustration** of single-modal attacks against fusion models for 3D object detection. $a_i$ denotes image features, $b_i$ Lidar features and $c_i$ fused features. Nodes marked in red denote affected features by perturbations on image.

*possible.* The degree of effect on the result depends on the weights of the image features (e.g., $w_{i,1}$) in the model. Larger weights could have severe consequence. In addition, the fusion approach could also affect the result. For example, if we only concatenate the second element ($a_2$) of $A$ with LiDAR features to calculate $C$ in the model, the previous adversarial attack on image cannot work anymore.

## E    GRADIENT-BASED ANALYSIS OF SENSITIVITY DISTRIBUTION RECOGNITION

Models employing different fusion approaches may exhibit varying distributions of vulnerable regions since the network design influences the training process and weights assignment. As a fusion model may have millions of weight parameters, identifying sensitive areas by weight analysis is infeasible. Hence, we proposed an automatic method to recognize the sensitivity distribution of a fusion model on single-modal input.

The main idea is to leverage the gradients of input data with respect to the adversarial loss. Larger gradients in an input region indicate that small perturbations have a higher impact on the adversarial goal, making it a more "sensitive" area. Taking Figure 9 as an example, through back-propagation, the calculation of an input pixel $x_i$'s gradient with respect to adversarial loss $L_{adv}$ is shown in Equation 13.

$$\nabla_{x_i} = \frac{\partial L_{adv}}{\partial x_i} = \sum_{j=1}^{2} \left( \frac{\partial L_{adv}}{\partial a_j} \frac{\partial a_j}{\partial x_i} \right)$$
$$= \sum_{j=1}^{2} \left[ \left( \sum_{k=1}^{3} \frac{\partial L_{adv}}{\partial c_k} \frac{\partial c_k}{\partial a_j} \right) \frac{\partial a_j}{\partial x_i} \right] \quad (13)$$
$$= \sum_{j=1}^{2} \left[ \left( \sum_{k=1}^{3} \frac{\partial L_{adv}}{\partial c_k} w_{k,j} \right) \frac{\partial a_j}{\partial x_i} \right]$$

As demonstrated, the weights of the single modality in fusion (i.e., $w_{k,j}(k = 1, 2, 3; j = 1, 2)$) are involved in the gradient calculation, and higher weights can lead to larger gradients. Therefore, we leverage the gradients as an overall indicator to understand the significance or vulnerability of different areas within the single-modal input.

Upon analyzing the optimization problem 1 from a gradient perspective, it becomes evident that the gradients of $M$ regarding the $L_{mask}$ steer towards the direction of minimizing mask values. In contrast, the gradients of $M$ with respect to $L_{adv}$ exhibit an opposite direction. As indicated in Equation 13, areas with higher sensitivity and greater weights in fusion possess larger gradients regarding $L_{adv}$, resulting in areas with higher intensity on the mask following optimization.

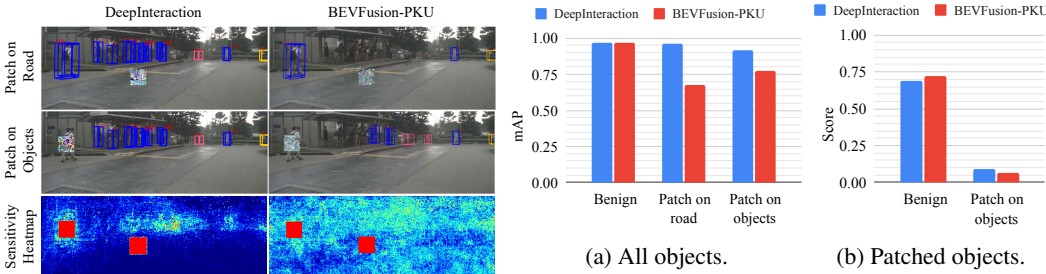

Figure 10: Property validation of the sensitivity heatmap using image-specific adversarial patches.

Figure 11: **Performance of image-specific adversarial patch** in validating properties of sensitivity heatmap.

## F  Model Selection Criteria

*1) Representativeness*: The selected models are the latest and most advanced fusion-based or camera-only models for 3D object detection. Published in 2022, each model's performance ranked top in the Nuscenes 3D object detection leaderboard (Motional, 2023). Additionally, each model employs the Transformer architecture (Vaswani et al., 2017) as the detection head, which is widely recognized as a cutting-edge design in object detection models and has been adopted in Tesla Autopilot (AIDay, 2022).

*2) Practicality*: The inputs to these models are multi-view images captured by six cameras surrounding a vehicle and the corresponding LiDAR point cloud collected by a 360-degree LiDAR sensor positioned on the vehicle's roof. This configuration of sensors is representative of practical autonomous driving systems and provides a more comprehensive sensing capability when compared to models that rely solely on front cameras (e.g., KITTI dataset (Geiger et al., 2012)).

*3) Accessibility*: All models are publicly available, ensuring that they can be easily accessed by researchers. The best-performing version of each model that utilizes camera-LiDAR fusion (or camera-only for BEVFormer) was selected and utilized in our experiments and can be found on their respective project repositories on GitHub.

Overall, these six models were selected as they provide a comprehensive and representative assessment of the latest advancements in camera-LiDAR fusion-based and camera-based 3D object detection and were deemed practical, accessible and relevant in the context of autonomous driving applications. The confidence (detection score) threshold for a valid bounding box detection is set to 0.3 in our evaluation to balance the false positive and false negative results.

## G  Property Validation of Sensitivity Heatmap

To validate the utility of sensitivity heatmaps in identifying vulnerable regions to adversarial attacks, we employ traditional image-specific adversarial patch on regions with distinct levels of sensitivity and evaluate the adversarial performance on two fusion models (i.e., DeepInteraction and BEVFusion-PKU) with distinct sensitivity types. We define a patch on both object and non-object areas to compare models' vulnerabilities since the two areas demonstrate different brightness on the sensitivity heatmap of DeepInteraction. More specifically, we define a patch area of size $50 \times 50$ on objects or roads within Scene 2, and generate the patch to minimize all object scores in the scene. We utilize the Adam optimizer with a learning rate of 0.001 and execute the optimization for 5000 iterations. Our experimental results are shown in Figure 10 and Figure 11.

In Figure 10, the first and second rows illustrate the patch on road and objects, respectively. The third row shows the sensitivity heatmaps with selected patch areas designated in red for reference purposes. Each column represents a distinct model. Our findings demonstrate that the effect of adversarial patches on the detection capability of DeepInteraction is negligible for the patch on the road; however, the patch on the object leads to compromised detection of the patched objects. These results align with the sensitivity heatmap, where the object area shows greater intensity compared to the road area, meaning the object area is more vulnerable. Contrarily, BEVFusion-PKU is globally

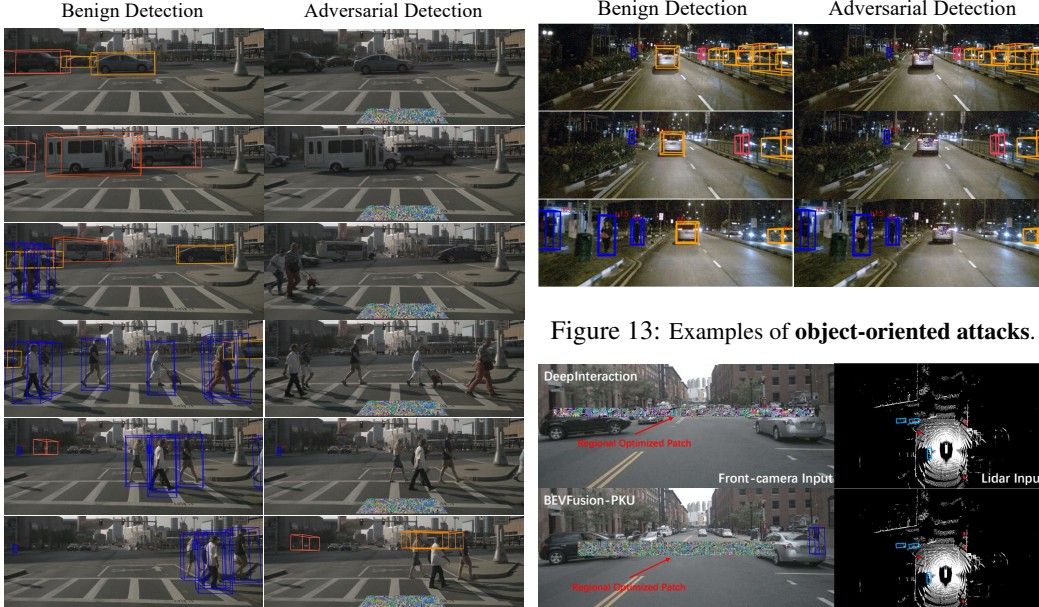

Benign Detection   Adversarial Detection     Benign Detection   Adversarial Detection

Figure 13: Examples of **object-oriented attacks**.

Figure 12: Comparison between the benign and adversarial cases of **scene-oriented attacks**.

Figure 14: Regional optimized adversarial patch with **one-stage optimization technique**.

sensitive, as shown by the sensitivity heatmap. It is observed that patches at both locations can cause false negative detection of surrounding objects, which is consistent with the heatmap and thus validates its accuracy.

Quantitative results are presented in Figure 11. We show in Figure 11a the average precision (AP) of the detection results of all pedestrians in the current scene, and we show in Figure 11b the average detection score of the two pedestrians covered by the patch in the patch-on-objects case. Our findings reveal that the patch on road only affects the BEVFusion-PKU model, and has no significant impact on DeepInteraction, as indicated by the unchanged AP. However, the patch on object is demonstrated to be effective for both models, as the detection scores for the patched objects decrease substantially in Figure 11b. Notably, the impact of the adversarial patch on DeepInteraction is confined mainly to the patched object, with only a minor effect on the AP of the scene (see the minor decrease in the third blue bar of Figure 11a). In contrast, the impact on BEVFusion-PKU is more comprehensive, affecting not only the patched object but also surrounding objects, leading to a greater decrease in AP than in DeepInteraction (see the third red bar of Figure 11a).

In summary, our findings substantiate the reliability of sensitivity heatmaps as an effective metric to identify susceptible regions and determine optimal targets for adversarial patch attacks. Our study has additionally revealed that models with global sensitivity are more susceptible to such attacks since both object and non-object areas can be targeted, while only object areas can be exploited in models with object sensitivity. Consequently, we have devised distinct attack strategies for models with different sensitivity distributions, which are described and assessed in Section 5.2 and Section 5.3.

## H  DETAILED EXPERIMENTAL SETUPS

**Training Devices and Computational Overhead.** Adversarial patches are generated utilizing a single GPU (Nvidia RTX A6000) equipped with a memory capacity of 48G, in conjunction with an Intel Xeon Silver 4214R CPU. On average, the generation of a sensitivity heatmap for a scene input $x$ for a specific fusion model requires approximately 1.2 hours, and we average across 10 unique scenes's sensitivity heatmap for the sensitivity type classification with Equation 4, hence the computational overhead for the first stage of our attack framework is around 12 hours. For

the second stage, around 3.1 hours are required to generate an effective adversarial patch for either object-oriented or scene-oriented attacks.

**Sensitivity Distribution Recognition.** During the optimization, we set the hyperparameters, in Equation 3 and Equation 1, $\lambda$ to 1, $s$ to 2, and $\gamma$ to 1. We adopt an Adam (Kingma & Ba, 2014) optimizer with a learning rate of 0.001 and conduct 2000 iterations of optimization. Due to the unique size of the input images $x$ for each model, we scale and crop the generated sensitivity heatmap of different models to 256×704 for better visualization.

**Scene-oriented Attacks.** We select one scene from the Nuscenes (Caesar et al., 2020) dataset in which the ego-vehicle is stationary at a traffic light (see Figure 12). Note that a Nuscenes scene represents a driving clip that lasts about one minute. This scene includes 490 frames of multi-modal data (multi-view images, 360-degree LiDAR point clouds, and object annotations), and the object types and locations vary across different frames. We split the scene into two subsets: the first 245 frames are used as the "training set" to generate an adversarial patch on the ground in front of the victim vehicle, using Equation 5. The remaining 245 frames are used as the "test set" to measure the attack performance. We report mean Average Precision (mAP) of seven categories of objects as the overall metric and average precision (AP) for each category as the specific metrics. The dimensions of the designated patch are 2 m×2.5 m, situated at a distance of 7 m from the ego vehicle on the ground. We project the patch region in the physical world onto the front-camera image with the LiDAR-to-image projection matrix obtained from the dataset. The optimization process utilizes the Adam algorithm, incorporating a batch size of 5 and a learning rate of 0.01, executed over 1000 iterations.

**Object-oriented Attacks.** We select an additional scene from the Nuscenes dataset, featuring a target vehicle driving in close proximity ahead of the ego-vehicle (see Figure 13). This scene comprises 260 sample frames, exhibiting dynamic variations in object types, positions, and background scenarios. Similar to the scene-oriented attacks, we utilize the first half of the frames as the "training set" and the latter half as the "testing set." We utilize the object-oriented projection (see Figure 4b) to map the patch image onto an area of the target object in the scene image, with physical location parameters $d$, $q$ and $\alpha$ extracted from the ground-truth 3D bounding boxes of the object. This area covers approximately one-ninth of the target vehicle's rear area with a size of 1 m × 1 m. Using Equation 5, the optimization process employs the Adam optimizer with a batch size of 5 and a learning rate of 0.01, executed for 1000 iterations. During testing, we measure and report the target object's average detection score and the mAP of detection results for other objects in the scene, including car, bus, pedestrian and motorcycle.

**Physical-world Experiments.** The image is captured using an iPhone 13, mounted on a camera tripod at a height of 1.5m, with a 70 degrees horizontal field-of-view (Allain, 2022) and a height aligned with the front camera configuration used in the Nuscenes dataset. Note that using a smartphone camera as our experimental equipment is not a compromised choice. Considering the overall cost and marginal performance improvement, cameras used on autonomous vehicles are usually not high-end ones and cameras on Tesla cost about 65$ each (Yoshida, 2020). In addition, the key sensor specifications (e.g., resolution and angular field-of-view) of our smartphone camera are kept consistent with the dataset sensors to achieve a faithful reproduction. Same as in the simulation, we begin by recording the intersection in a benign scenario, generate the scene-oriented adversarial patch targeting BEVFusion-PKU and print it. The printed patch measures 2m×2.5m. We ask three people to stand at locations similar to those of pedestrians in the original dataset image, capturing benign and adversarial data with and without the deployed patch for validation.

# I   QUALITATIVE RESULTS

**Scene-oriented Attacks.** Figure 12 demonstrates the benign and adversarial scenarios in the test set and the corresponding object detection outcomes of BEVFusion-PKU (Liang et al., 2022). It is evident from the right column that the majority of objects in proximity to the adversarial patch are undetected, encompassing cars, buses, pedestrians, and other entities, which highlights a considerable safety concern for both the ego-vehicle and individuals at the intersection.

**Object-oriented Attacks.** Figure 13 demonstrates the patched target vehicle in multiple frames and the failure of detecting it with DeepInteraction as the subject fusion model.

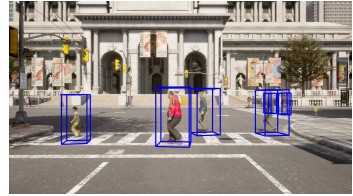 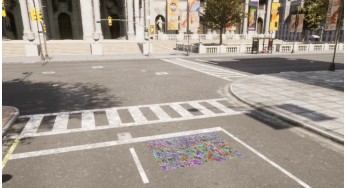 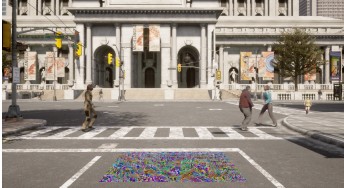

(a) Benign Scenario        (b) Patch on the Ground        (c) Adversarial Scenario

Figure 15: Attacks in CARLA.

Table 5: Attack performance at different times of the day with various lighting conditions.

| Time of the day | Benign mAP | Adversarial mAP | Difference ↑ |
|---|---|---|---|
| 9:00 AM | 0.428 | 0.191 | 55.37% |
| 12:00 PM | 0.457 | 0.184 | 59.74% |
| 3:00 PM | 0.481 | 0.165 | 65.70% |
| 6:00 PM | 0.43 | 0.204 | 52.56% |

## J    COMPARISON WITH PATCH ATTACK AGAINST CAMERA-ONLY MODELS

We compare our two-stage attack framework with the state-of-the-art one-stage patch optimization technique that targets camera-only models (Cheng et al., 2022). This method, originally designed to attack monocular depth estimation models with a regionally optimized patch on the target object, employs a differentiable representation of a rectangular patch mask, parameterized by four border parameters, to simultaneously optimize both patch region and content. Consequently, it can pinpoint the most sensitive area for maximizing the attack impact. In our scenario, we optimize the patch region and content over the entire scene with the objective of false negative detection of objects. The results of incorporating this technique in our attack against fusion models are illustrated in Figure 14. As shown, the patches in Figure 14, using the same number of perturbed pixels as in Figure 2, could undermine detection results for DeepInteraction and BEVFusion-PKU on a single-image. These findings corroborate the efficacy of the patch optimization technique (Cheng et al., 2022) on individual images. Nevertheless, the resulting patch presents significant limitations due to its impracticality in real-world contexts. Its broad reach, which includes both dynamic entities and static backgrounds, renders it unsuitable for physical deployment and thereby undermines its overall applicability. Moreover, the variability of background scenes and objects across different image frames imposes significant fluctuations on the optimal patch region as determined by this technique, thereby undermining its generality. In Cheng et al. (2022), the patch region is narrowly constrained to a specific target object, as opposed to optimizing across the entire scene in our context. This circumvents the aforementioned limitations. In contrast, our two-stage approach decouples the location and content optimization, and it is guaranteed to generate both deployable and effective patches for any scene (see Figure 12 and Figure 13). Consequently, our proposed two-stage method emerges as a more universal framework, with the potential to integrate prior methods into its second stage for refinement of the patch region following the determination of the vulnerable regions and attack strategies.

## K    ADDITIONAL PRACTICALITY VALIDATION

To gather data in a format similar to the Nuscenes dataset, we utilize the CARLA (Dosovitskiy et al., 2017) simulator and calibrate our sensors to match the Nuscenes configuration. The collected data is organized to match the format of the Nuscenes dataset, allowing the fusion models to process the simulator data without retraining. As shown in Figure 15a, we first collect benign data, in which the ego-vehicle stops at an intersection and the surrounding pedestrians can be detected correctly. Subsequently, the scene-oriented adversarial patch for the intersection is generated using the gathered data, as outlined in Section 5.2. We use BEVFusion-PKU as the subject fusion model. This patch, of dimensions 2m × 2.5m and situated 7m from the ego-vehicle, is then integrated into the simu-

Table 6: Physical attack performance under various lighting conditions.

| Time | Benign Score | Adversarial Score | Difference ↑ |
|------|------|------|------|
| 9:00 AM | 0.663 | 0.253 | 61.84% |
| 6:00 PM | 0.672 | 0.217 | 67.71% |

Table 7: Attack performance with **various patch distances**. Metric: The % of AP degradation under attack.

Table 8: Attack performance with **various patch angles.** Metric: The % of AP degradation under attack.

| Distance | mAP | Automotive | Barrier | Pedestrain | Bicycle |
|------|------|------|------|------|------|
| 7.0m | 64.77% | 31.70% | 70.04% | 48.08% | 100.00% |
| 7.5m | 62.53% | 54.01% | 45.43% | 61.52% | 85.05% |
| 8.0m | 70.49% | 83.92% | 49.33% | 75.76% | 74.95% |
| 8.5m | 71.25% | 3.26% | 98.44% | 65.76% | 100.00% |
| 9.0m | 59.17% | 0.18% | 68.49% | 51.72% | 100.00% |
| 9.5m | 8.56% | 1.97% | 16.82% | 17.07% | 0.00% |
| 10.0m | 0.27% | 0.04% | 0.00% | 1.01% | 0.00% |
| Average | 56.13% | 29.17% | 58.09% | 53.32% | 76.67% |

| Angles | mAP | Automotive | Barrier | Pedestrain | Bicycle |
|------|------|------|------|------|------|
| -15 | 49.92% | 45.45% | 47.88% | 76.87% | 27.98% |
| -10 | 68.13% | 6.77% | 88.53% | 61.01% | 100.00% |
| -5 | 49.31% | 44.73% | 49.78% | 76.46% | 24.95% |
| 0 | 70.49% | 83.92% | 49.33% | 75.76% | 74.95% |
| 5 | 65.65% | 0.82% | 89.53% | 55.35% | 100.00% |
| 10 | 48.39% | 43.62% | 46.88% | 76.46% | 25.05% |
| 15 | 69.20% | 2.11% | 82.52% | 73.64% | 100.00% |
| Average | 60.16% | 32.49% | 64.92% | 70.79% | 64.70% |

lation environment, at the intersection, as depicted in Figure 15b. As the ego vehicle approaches the patch in the adversarial scenario (See Figure 15c), the fusion models fail to detect surrounding pedestrians. The AP of the model's detection performance for pedestrians has dropped by 59%, from 0.457 in the benign case to 0.184 in the adversarial case. Videos of the simulation can be found at `https://youtu.be/xhXtzDezeaM`. Note that the benign data used for patch generation incorporates different pedestrians from the adversarial scenario, thus underscoring the patch's generality.

To further validate the robustness of our attacks in different lightning conditions, we repeated the above experiment at different times throughout the day to account for changes in light direction and intensity. The results are shown in Table 5. The first column denotes the time we conduct the experiment in simulation. The second and third columns indicate the benign and adversarial performance of BEVFusion-PKU respectively. The fourth column presents the percentage of performance degradation by comparing the benign and adversarial performance. As shown, our patch remains a robust attack performance at different lighting conditions. To evaluate the impact of varying lighting conditions in the physical world, we also conduct the physical-world experiments at different times, results can be found in Table 6. As shown, at different times of the day, the average detection scores of the pedestrians are significantly decreased after the patch is deployed, which validates the physical robustness of our patch under different lighting conditions.

## L   VARYING DISTANCE AND VIEWING ANGLES

In order to conduct a comprehensive evaluation of our adversarial attacks, we modify the distance and viewing angles of the adversarial patch and assess the scene-oriented attack performance on the BEVFusion-PKU(Liang et al., 2022) model. During the distance evaluation, we position a patch with dimensions of 3m × 5m on the ground in front of the ego-vehicle, varying the distance from 7 meters to 10 meters. In the viewing angle assessment, we place the patch 8 meters from the ego-vehicle and rotate it around the z-axis (i.e., the vertical axis perpendicular to the ground) from -15 degrees to 15 degrees. All other experimental settings remain consistent with Section 5.2. The performance degradation caused by our attack is reported in Table 7 and Table 8. The first column represents various distances and viewing angles, while the second column exhibits the percentage of mAP degradation of detection results. Subsequent columns display the specific average precision degradation for individual object categories. The automotive group encompasses cars, trucks, buses, and trailers. As illustrated in Table 7, the adversarial patch demonstrates strong attack performance within the range of 7 meters to 9 meters, with diminishing effectiveness beyond 9 meters. This decrease in efficacy may be attributed to the patch appearing smaller in the camera's field of view as the distance increases, resulting in fewer perturbed pixels and a consequent decline in attack performance. In real-world scenarios, distances less than 9 meters are practical for initiating

Table 9: Detection score of the target object under different distances and patch sizes.

| Distance | Ben. | 1m×1m | | 1m×0.5m | | 0.5m×0.5m | |
|---|---|---|---|---|---|---|---|
| | | Adv. | Diff.↑ | Adv. | Diff.↑ | Adv. | Diff.↑ |
| 5m | 0.711 | 0.074 | 89.59% | 0.081 | 88.61% | 0.146 | 79.47% |
| 6m | 0.726 | 0.112 | 84.57% | 0.108 | 85.12% | 0.144 | 80.17% |
| 8m | 0.682 | 0.105 | 84.60% | 0.137 | 79.91% | 0.191 | 71.99% |
| 10m | 0.643 | 0.127 | 80.25% | 0.242 | 62.36% | 0.402 | 37.48% |
| 15m | 0.655 | 0.146 | 77.71% | 0.48 | 26.72% | 0.626 | 4.43% |

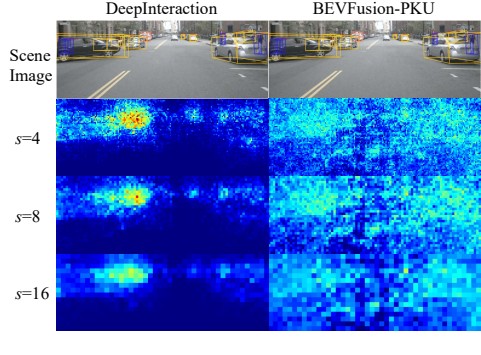

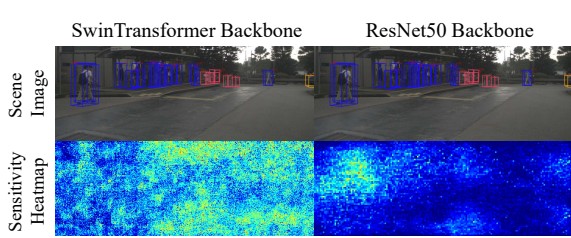

Figure 17: Sensitivity heatmap of BEVFusion-MIT(Liu et al., 2023b) using **different image backbones**.

Figure 16: Sensitivity heatmap with **different granularity**.

scene-oriented attacks. For example, street paint at intersections is typically less than 9 meters to the leading vehicle. Observations from Table 8 indicate that the patch maintains robust attack performance across varying angles, with optimal performance occurring at no rotation (0°). Both distance and angle results reveal that attack performance is better for foreground objects (e.g., pedestrians and bicycles) situated closer to the adversarial patch and ego-vehicle.

We also conduct object-oriented attacks with different patch sizes and shapes at various distances. We use the CARLA simulator to collect data with the target object locating at a range of distances from the ego-vehicle. Then we generate an object-oriented adversarial patch against the target object following Equation 5. The subject fusion model is DeepInteraction and we test the attack performance on another scene collected from the simulator. Experimental results are presented in Table 9. The first column denotes the distance with the target vehicle. The second column reports the benign detection score of the target object without patch being attached. The following columns show the adversarial detection scores and the difference with benign case in percentage after the patch is applied. Larger difference indicates higher performance degradation and better attack performance. Patches of three difference sizes are evaluated and results show that the minimum patch size necessary for a successful attack is contingent upon the distance between the target object and the victim vehicle. The critical factor is the pixel count influenced by the patch in the input images, and a smaller patch can exhibit efficacy when in close proximity to the victim vehicle. The 0.5m × 0.5m patch still performs well at a closer distance (e.g. less than 8m). Moreover, the patch does not need to be in square shape. Attackers have the flexibility to define patches with rectangular or arbitrary shapes to circumvent covering the target vehicle's license plate.

## M  VARYING GRANULARITY OF SENSITIVITY HEATMAP

Our sensitivity recognition algorithm employs the hyper-parameter $s$ in Equation 1 to regulate the granularity of the mask $M$ applied to the perturbation, which denotes the size of a unit area. In order to assess whether the distribution of sensitivity would be influenced by variations in mask granularity, we perform experiments with diverse settings of $s$. By default, we establish $s$ as 2, and the corresponding outcomes are illustrated in Figure 5. Outcomes derived from different granularity settings are depicted in Figure 16. We assign values of 4, 8, and 16 to $s$, and generate the sensitivity heatmap for DeepInteraction and BEVFusion-PKU. The first row in Figure 16 exhibits the origi-

nal scene image and ground-truth objects. As the results indicate, the sensitivity distribution for a given scene remains generally consistent across differing granularity of masks, and the two types of sensitivity continue to exhibit unique attributes. DeepInteraction retains its object-sensitive nature, whereas BEVFusion-PKU persistently demonstrates global sensitivity.

## N   DEFENSE DISCUSSION

**Architecture-level defense.** Our analysis reveals that camera-LiDAR fusion models exhibit two types of sensitivity to adversarial attacks: global and object sensitivity. Globally sensitive models are more vulnerable as they are susceptible to both scene-oriented and object-oriented attacks. In contrast, object-sensitive models are more robust due to their smaller sensitive regions and resistance to non-object area attacks. Both model types, however, perform similarly in benign object detection. We investigate the architectural designs to understand the cause of different sensitivity types. We find that object-sensitive models (DeepInteraction (Yang et al., 2022), UVTR (Li et al., 2022b), Transfusion (Bai et al., 2022) and BEVFormer (Li et al., 2022c)) employ CNN-based ResNet (He et al., 2016) as their image backbone, while globally sensitive models (BEVFusion-PKU (Liang et al., 2022) and BEVFusion-MIT (Liu et al., 2023b)) use transformer-based SwinTransformer (Liu et al., 2021b). To further investigate, we retrain BEVFusion-MIT (Liu et al., 2023b) with ResNet50 and compare the sensitivity heatmap to the original SwinTransformer model, as shown in Figure 17. The results indicate that sensitive regions are more focused on objects when using ResNet, suggesting that the image backbone significantly impacts model vulnerability. An explanation is that the CNN-based ResNet focuses more on local features due to its small convolutional kernels (usually $3\times3$ pixels), while the transformer-based SwinTransformer captures more global information through self-attention mechanism. Consequently, adversarial patches distant from objects can still affect detection in transformer-based backbones. To enhance the model's security against such attacks, incorporating ResNet50 as the image backbone is a preferable architectural choice.

## O   LIMITATIONS

While our results demonstrate successful attacks against state-of-the-art camera-LiDAR fusion models using the camera modality, we have not conducted an end-to-end evaluation on an actual AV to illustrate the catastrophic attack outcomes (e.g., collisions or sudden stops). This limitation stems from cost and safety concerns and is shared by other studies in AV security research (Cao et al., 2021; 2019; Sato et al., 2021). It should be noted that the most advanced fusion models examined in this work have not yet been implemented in open-sourced production-grade autonomous driving systems. Publicly available systems, such as Baidu Apollo (BaiduApollo) and Autoware.ai (Autoware), employ decision-level fusion rather than the advanced feature-level or data-level fusion here. As a result, we did not demonstrate an end-to-end attack in simulation. However, feature-level fusion is gaining attraction in both academia (Liu et al., 2023b; Liang et al., 2022; Yang et al., 2022) and industry (Li et al., 2022a), driven by advancements in network designs and enhanced performance. Our technique is applicable to all feature-level and data-level fusion models. Furthermore, our evaluations on a real-world dataset, in simulation, and in the physical world, using industrial-grade AV sensor array underscore the practicality of our attack.

## P   THREAT MODEL

Our attack assumes that the attacker has complete knowledge of the camera-LiDAR fusion model used by the target autonomous driving vehicle. Therefore, our attack model is considered to be in a white-box setting. This assumption is congruent with analogous endeavors in the literature dedicated to adversarial attacks on autonomous driving systems (Cao et al., 2021; Sato et al., 2021; Zhang et al., 2022; Zhu et al., 2023; Jin et al., 2023). To achieve this level of knowledge, the attacker may employ methodologies such as reverse engineering the perception system of the victim vehicle (Lambert, 2021), employing open-sourced systems, or exploiting information leaks from insiders. In addition, the attacker is clear about the target object and driving scenes he wants to attack. For example, he can record both video and LiDAR data using his own vehicle and devices while following a target object (for object-oriented attacks) or stopping at a target scene (for scene-oriented attacks). Leveraging this pre-recorded data, the attacker undertakes a one-time effort to

generate an adversarial patch using our proposed approach. The attacker can then print and deploy the generated patch on the target object or in the target scene for real-time attacks against victim vehicles in the physical world.

## Q   DISCUSSION ON AUTONOMOUS VEHICLE SECURITY

While some production-grade Autonomous Vehicles (AVs) employ entirely vision-based perception algorithms, such as Tesla, the majority utilize camera-LiDAR fusion techniques (e.g., Baidu Apollo (BaiduApollo), Google Waymo Waymo and Momenta (Momenta)). The focus of this research is on 3D object detection, a crucial task in AV perception that allows the AV to understand surrounding obstacles and navigate safely within the physical environment. The attacks proposed in this study against advanced fusion models challenge the security assumption of multi-sensor fusion-based perception algorithms, particularly the widely used early-fusion scheme (i.e., data-level and feature-level fusion strategies). Given that the adversarial patch we generated impacts object detection in each data frame, subsequent tasks such as motion tracking and trajectory prediction within the victim vehicle will also be compromised. This increases the likelihood of the victim vehicle colliding with undetected targets, potentially resulting in life-threatening situations. It is noteworthy that the attack performance intensifies as the distance between the affected vehicle and the adversarial patch decreases, implying that the driver may not have adequate reaction time, leading to a braking distance that exceeds the actual distance.

The conduct of the planning module on the victim vehicle depends on the speed of the victim vehicle and its relative speed in relation to the target vehicle, as well as surrounding environments. When the two speeds are relatively low, the victim vehicle may not necessarily change lanes or brake in advance before the target vehicle approaches within a certain distance. For instance, in scenarios of traffic congestion, the victim vehicle may follow the target vehicle in front at a close distance (e.g., less than 8 meters) due to their shared low speed. Moreover, in intersections or parking lots, the victim vehicle may halt behind the target vehicle, thus a failure in detection could result in a collision when the victim vehicle starts.

Given that our attacks are executed in a white-box setting, it is possible that the AD industry could enhance its security mechanisms to prevent model leakage. We anticipate that our investigation into the single-modal robustness of fusion models, along with the proposed attack methods, will attract the attention of developers and engineers within the AD industry. This could potentially stimulate the creation of more robust perception modules and autonomous driving systems.

