# OpenReview forum: "Fusion Is Not Enough: Single Modal Attacks on Fusion Models for 3D Object Detection"
_ICLR.cc/2024/Conference — ICLR 2024 poster_

### Official Review · Reviewer_nx8r · 2023-10-27

**Soundness:** 3 good
**Presentation:** 3 good
**Contribution:** 3 good
**Rating:** 8
**Confidence:** 4

**Summary:**

The author proposes a two-stage optimization strategy for camera-LiDAR fusion models via using adversarial patches in camera images. The paper explores single-modal attack on fusion models, an interesting yet under-explored problem.

**Strengths:**

* The paper is well-written and explores an interesting topic.
* The proposed method can decrease the model performance by a large margin, showing the possibility of dramatically deteriorating the system performance by only attacking single-modality.

**Weaknesses:**

* The effectiveness of the proposed two-stage optimization approach needs further justifications. Only showing the performance drop on fusion models is not enough. Comparisons with other single-stage attacks are also needed to demonstrate the effectiveness. Without proper benchmarks and comparisons with other SOTA algorithms, it is hard to justify the effectiveness of the technical contributions.
* How to ensure the feasibility of the adversarial patches? Since the gradient optimization may find patches in the undeployable areas e.g., sky, can the proposed approach ensure the attack is feasible in the real physical world? Also in the paper, the author assumes the lidar data would not be changed. Since the patch may influence the lidar intensity or introduce extra points, please provide justifications for this assumption.

**Questions:**

* What is the sensitivity of single-modal methods vs multi-modal (LiDAR/camera) methods?

---

> ### Author Response · Authors · 2023-11-18
> **Official Response to Reviewer nx8r**
>
> We express our deepest gratitude for the reviewer’s time and astute comments. It is our pleasure to address the primary concerns in the following sections.
>
> **Q1: The effectiveness of the proposed two-stage optimization approach needs further justifications.**
>
> **A1**: Thank you sincerely for providing valuable insights.  In our original manuscript, we took care to include a comprehensive comparative analysis, particularly with respect to other single-stage attacks, notably the state-of-the-art (SOTA) one-stage camera-only patch region optimization methods discussed in Appendix J. The results of this comparative study underscore the limitations of the SOTA method in generating physically deployable patches when considering the entire scene.  Additionally, in the motivation section, we presented an evaluation of traditional single-stage patch attacks, emphasizing their failure in achieving success if the predefined patch region does not align with a vulnerable region for the subject fusion model. Our two-stage attack framework, through its adept identification of vulnerable regions and utilization of distinct attack strategies based on sensitivity types, aims to generate both deployable and effective adversarial patches. This, we believe, substantiates the efficacy and versatility of our proposed method.
>
> **Q2: How to ensure the feasibility of the adversarial patches?**
>
> **A2**: Your thoughtful question is much appreciated. In contrast to the SOTA one-stage gradient-based patch optimization approach (as shown in appendix J), our two-stage approach has solved the problem of undeployable adversarial patches. Notably, the patch's location in our approach is not optimized with gradients concerning the entire scene; rather, it is determined by highly practical projection functions. The optimization with gradients in the second stage is exclusively applied to the patch texture. As detailed in Section 4, following the recognition of sensitivity types of the fusion model in the first stage, different attack strategies are deployed in the second stage based on sensitivity types. The patch's location on the image is contingent on the projection function (i.e., proj_x()) in specific attack strategies (See Figure 4). For scene-oriented attacks, we project the patch onto a predefined physical location on the ground, and for object-oriented attacks, we dynamically project the patch onto a target object using reverse-engineered projection parameters based on different 3D locations and orientations of the object from the driving footage. Consequently, the patch generated by our method is highly deployable and feasible in the real world.
>
> We posit a reasonable assumption that our patch does not affect LiDAR since it is designed to be affixed to the ground (for scene-oriented attacks) or a target vehicle (for object-oriented attacks) during deployment. Our primary alteration occurs in the surface texture of the 3D scene, preserving the 3D structure captured by the LiDAR sensor. The material of the patch (e.g., paper) is common and has little influence on the LiDAR intensity. Hence the impact of our approach on the LiDAR data is minimized and it is also difficult to recognize the adversarial information using only the LiDAR sensor.
>
>
> **Q3: What is the sensitivity of single-modal methods vs multi-modal (LiDAR/camera) methods?**
>
> **A3**: Your question is both thoughtful and appreciated. Our attack framework extends its applicability to single-modal methods, as evidenced by our evaluation with a camera-only model (BEVFormer) in the original manuscript. The last row of Figure 5 shows the sensitivity heatmap of BEVFormer, which demonstrates higher sensitivity in object areas and the model is classified as object sensitivity. Similar to other object-sensitive models, BEVFormer is robust to scene-oriented attacks and vulnerable to object oriented attacks as evidenced in Table 1 and Table 2. Compared with multi-modal models, the single-modal model exhibits  worse benign performance and experiences more pronounced performance degradation under object-oriented attacks.

---

> ### Comment · Reviewer_nx8r · 2023-11-21
>
> Thanks for the detailed feedback.
>
> Q1: Appendix J only shows the qualitative visualizations while quantitative results are needed to justify the performance boost.
>
> Q2: How to ensure the projection always projects to the ground?

---

> ### Author Response · Authors · 2023-11-21
> **Official Response to the Follow-up Question**
>
> Thanks a lot for your continuous help in improving our paper. We answer your new questions below.
>
>
> **Q1: quantitative results are needed to justify the performance boost.**
>
>
> **A1**: Your thoughtful comments are appreciated. Allow us to clarify that the contribution of our approach is not the boost of attack performance in the digital space, but the practicality and effectiveness of the patch attack in the physical world. The qualitative visualization in appendix J demonstrates that the patch generated by the SOTA method, which encompasses a one-stage optimization of the patch’s content and location, spans over multiple physical objects and is not deployable. In contrast, our two-stage approach decouples the location and content optimization, and it is guaranteed to generate both deployable and effective patches for any scene (see Figure 12 and Figure 13).  Hence our approach is more versatile and universal.
>
>
> **Q2: How to ensure the projection always projects to the ground?**
>
>
> **A2**: For the scene–oriented attacks, as shown in Figure 4a, we pre-define the 3D location of the patch with parameters like height $g$, distance $d$ and viewing angle $\alpha$, in the coordinate system of the ego-vehicle's front camera. Then we project the 3D coordinates of the patch back onto the 2D scene image (step 2). Given that we know the camera's height $g$ above ground from the dataset, we ensure by definition that the patch is positioned on the ground when using this camera height.

---

> ### Comment · Reviewer_nx8r · 2023-11-22
>
> Thanks for the detailed comments and the great efforts in rebuttal. All my questions are addressed. Considering the novel physically plausible attack methods and technical contributions for multi-modal sensor fusion adversarial attack, I would like to increase the ratings. The analysis and additional explanations in the rebuttal should be added in the final manuscript for better readability. Thank you!

---

> > ### Author Response · Authors · 2023-11-22
> > **Thank You for the Feedback**
> >
> > We express our gratitude for your valuable feedback and reevaluation of our contribution. We have updated the additional analysis and explanations in the latest version.  Thanks a lot for your continuous help in making our manuscript better!

---

### Official Review · Reviewer_DXNR · 2023-10-28

**Soundness:** 3 good
**Presentation:** 2 fair
**Contribution:** 2 fair
**Rating:** 6
**Confidence:** 3

**Summary:**

This paper proposes an adversarial attack method, which could attack multi-modal 3D object detection methods from the image input, making the attack easy to implement.
It first recognizes the proper attack type and proposes different methods for each type.
Experiments are conducted with multiple popular multi-modal detectors, showing promising performance.

**Strengths:**

- The motivation that attacks from image input is reasonable, which is practical real world.
- The proposed method overall makes sense.
- Authors provide a demo, which makes the application of the proposed method more clear.
- Authors conducted extensive experiments with many SOTA the art detectors, making the results convincing.

**Weaknesses:**

- Attack from only the image side has limited application. In particular, some methods do not conduct feature-level fusion between image and LiDAR like [1][2].  The two modalities are decoupled in these methods. Even if the image modality is totally failed, they can output reasonable results.
- The writing is not clear, especially in page 4 and page 5. There are very long paragraphs and many notations without clear organization, making it hard to follow the detailed method. I list some detailed questions in the following question box.
- The method is not well-motivated. The proposed method seems to be a general attacking method. Are there any special designs to solve problems in image-only attacking or autonomous driving scenes? What is the difficulty of image-only attacks?

[1] Frustum PointNets for 3D Object Detection from RGB-D Data \
[2] Fully Sparse Fusion for 3D Object Detection

**Questions:**

- In attack strategies, is the noise patch shared by the whole dataset? I saw the patch keeping changing in object-level attacks but remaining unchanged in scene-level attacks.
- Is the mask in Sensitivity Distribution Recognition shared by the whole dataset?
- I do not understand the form of the proj_x in Eq. 6 and why it is necessary.

---

> ### Author Response · Authors · 2023-11-18
> **[Part 1/2] Official Response to Reviewer DXNR (Q1, Q2 & Q3)**
>
> We express our deepest gratitude for the reviewer's invaluable time and astute comments. We have made every effort to address the principal concerns as follows, and we sincerely appreciate the opportunity to enhance our work through your insightful feedback.
>
> **Q1: Some methods do not conduct feature-level fusion between image and LiDAR. The two modalities are decoupled in these methods.**
>
> **A1**: We appreciate your insightful remarks. The methods you have brought up are in the category of data-level fusion. As expounded in Appendix B of our manuscript, for the data-level fusion strategy, some studies use the extracted image information to augment the LiDAR points. In the evaluation section, we demonstrated the effectiveness of our proposed attack on a data-level fusion model (i.e., PointAugmenting) that utilizes image features extracted by CNN to hence LiDAR data. In the first study you mentioned, the authors employ CNN-based 2D object detection to identify object regions in the image input, then use the extruded 3D viewing frustum to extract corresponding points from the LiDAR point cloud for bounding box regression. In the second study, the authors use semantic segmentation on the image input to generate the frustum. Both methods depend on information from the camera input (i.e., 2D object detection or semantic segmentation results) to augment point cloud-based object detection. Therefore, adversarial attacks on images can deceive  object detection or semantic segmentation as in [1-3], leading to the failure of the extruded 3D frustum in capturing the corresponding LiDAR points of each object. This would subsequently affect the 3D bounding box regression. Your insights are greatly appreciated, and a discussion on these studies has been included in Appendix B, with appropriate citations.
>
> **Q2: Is the noise patch shared by the whole dataset?**
>
> **A2**:  Thank you for your comments and queries. The adversarial patch is shared across various frames in the dataset, with each patch optimized for a scene (in scene-oriented attacks) or a target object (in object-oriented attacks). In the context of object-oriented attack strategies, we project the patch image onto the rear of the target vehicle in each scene image using varying projection parameters based on the target object's location in 3D space (see Figure 4b). The varying patch appearances in object-oriented attacks result from the changing projection parameters and the interpolation among pixels in each data frame.  We have included a detailed threat model in Appendix Q of our revised manuscript.
>
> **Q3: Is the mask in Sensitivity Distribution Recognition shared by the whole dataset?**
>
> **A3**: No, the mask is not shared. Each mask (i.e., sensitivity heatmap) is defined and optimized for a specific frame of data extracted from the dataset. For the sensitivity type classification, as per Equation 4, the average sensitivity of object areas and non-object areas is calculated across various data frame $x$’s sensitivity heatmaps. We have refined the description and notations in the revised version for clarity.

---

> ### Author Response · Authors · 2023-11-18
> **[Part 2/2] Official Response to Reviewer DXNR (Q4 & Q5)**
>
> **Q4: I do not understand the form of the proj_x in Eq. 6 and why it is necessary.**
>
> **A4**: proj_x() signifies the synthesis of the original patch image (see the “2D patch image” in Figure 4a) onto a specific area of the scene image (see the “captured image” in the center of Figure 4a)  to simulate how the patch would look once it's physically deployed. This projection is necessary for enhancing the physical-world robustness of the generated patch and minimizing the disparity between digital space and the physical world. As outlined, equations 7-9 provide an expanded representation, where ($u^p$, $v^p$) denotes the coordinates of a pixel on the original patch image, and ($u^s$, $v^s$) represents the pixel's corresponding coordinates on the scene image. proj_x() establishes this mathematical connection based on projection parameters (e.g., lateral and longitudinal distance, and viewing angle of the patch in the physical world). Specifically, for object-oriented attacks, as the target object’s location and orientation vary across data frames, the patch would move with the target object once deployed physically. Hence, the patch’s projection parameters should differ across frames for a faithful simulation during patch training before deployment. We innovatively extracted those parameters dynamically from the detected 3D position of the target object as we stated in our methods. We have revised the presentation of our method in the updated manuscript to make it more clear.
>
> **Q5: Are there any special designs to solve problems in image-only attacking or autonomous driving scenes? What is the difficulty of image-only attacks?**
>
> **A5**: We acknowledge your concerns and would like to highlight the unique challenges inherent in image-only attacks that have guided our method's unique design. Most prominently, depending on model architectures and fusion strategies, the vulnerable regions change across different fusion models. As a result, directly attacking a specific area, akin to the SOTA patch attack [4], may fail when the targeted area is not a vulnerable region (e.g., attacking the road for object-sensitive models, as depicted in Figure 2). Hence, a unique challenge is to determine the vulnerable region, and then the corresponding attack strategy can be employed. Our sensitivity distribution recognition algorithm addresses this challenge. A straightforward extension to the SOTA attack would be to directly optimize some patch on entire scene images. However, the generated patch may span over multiple physical objects and is not deployable (see Appendix J). Our two-stage design ensures physical deployability, and its compatibility with SOTA camera attacks reinforces it as a more universal solution. Rather than viewing the generality of our attack framework as a limitation, we consider it a strength. Another addressed challenge is reverse engineering precise projection functions for different target object locations and orientations from driving footage, facilitating faithful scene synthesis during offline patch generation (as elucidated in Q4).
>
> **Reference**
>
> [1] Xie, Cihang, Jianyu Wang, Zhishuai Zhang, Yuyin Zhou, Lingxi Xie, and Alan Yuille. "Adversarial examples for semantic segmentation and object detection." In ICCV, 2017.
>
> [2] Huang, Lifeng, Chengying Gao, Yuyin Zhou, Cihang Xie, Alan L. Yuille, Changqing Zou, and Ning Liu. "Universal physical camouflage attacks on object detectors." In CVPR, 2020.
>
> [3] Nesti, Federico, Giulio Rossolini, Saasha Nair, Alessandro Biondi, and Giorgio Buttazzo. "Evaluating the robustness of semantic segmentation for autonomous driving against real-world adversarial patch attacks." In WACV, 2022.
>
> [4] Cheng, Zhiyuan, James Liang, Hongjun Choi, Guanhong Tao, Zhiwen Cao, Dongfang Liu, and Xiangyu Zhang. "Physical attack on monocular depth estimation with optimal adversarial patches." In ECCV, 2022.

---

> > ### Comment · Reviewer_DXNR · 2023-11-20
> > **Follow-up question to Q5**
> >
> > I appreciate the authors' effort in the rebuttal, which addresses some of my concerns. However, I am still confused about Q5.
> >
> > - In the rebuttal, the authors said "As a result, directly attacking a specific area may fail when the targeted area is not a vulnerable region". However, from my personal view, it holds true for all attacking methods, not only the image-only attacking. Thus, it is not a challenge of image-only attack.
> >
> > - It remains unclear what the type of the proposed method is.  It is a general method to attack fusion models or a specific method for image-only attacking?  If it is a specific model, the unique designs for the image-only setting are not very clear, because I don't think the sensitivity recognition is rooted in the image-only setting, as mentioned above. If it is a general model, why do the authors emphasize the image-only setting? I encourage the authors to make it more clear, otherwise, the logic of this paper is not firmed.

---

> > > ### Author Response · Authors · 2023-11-20
> > > **Official Response to the Follow-up Question**
> > >
> > > We express our gratitude to the time and efforts you have dedicated to our rebuttal. We address the new concerns as follows.
> > >
> > > **Q1: Is it a general method to attack fusion models or a specific method for image-only attacking?**
> > >
> > > **A1**: Your thoughtful comments are appreciated. Our attack is a specific method for image-only attacking against fusion models. As described in Introduction, previous LiDAR-related attacks pose implementation challenges, necessitating additional equipment like photodiodes, laser diodes, or industrial-grade 3D printers to manipulate LiDAR data, thereby increasing deployment costs for attackers. Our image-only attacks, in comparison, are more affordable and easier to deploy, enabling attackers to print generated adversarial patches with home printers. This reduction in effort heightens the threat to autonomous vehicle security.
> > >
> > > It is possible that we may have misinterpreted the generality you referred to in your initial question. Our method is not intended to be general in attacking every modality in fusion-based models, and our primary focus is the camera modality. The generality of our work is that our image-only attack is not confined to fusion-based models, but is also applicable to image-based models (e.g., BEVFormer in our Evaluation).
> > >
> > > **Q2: The unique designs for the image-only setting are not very clear, because I don't think the sensitivity recognition is rooted in the image-only setting.**
> > >
> > > **A2**: We sincerely appreciate your inquiries. Although the sensitivity recognition problem is not just rooted in the image-only setting, our proposed method addresses this issue from an image-centric standpoint, thereby facilitating the more effective and practical image-only attacks.  Our proposed method innovatively utilizes a dual optimization of perturbation and perturbation masks on the input image and visualizes the converged mask to achieve sensitivity heatmap. Compared with the SOTA patch attack that can be adapted to tackle this challenge on images (see appendix J), our two-stage design guarantees physical deployability and provides a more comprehensive assessment of the vulnerable regions for image-only attacks. Thanks for your insightful comments, we have highlighted our logic and motivation in the updated version of the manuscript.

---

> > > > ### Author Response · Authors · 2023-11-21
> > > > **Message from Authors**
> > > >
> > > > Dear reviewer DXNR,
> > > >
> > > > Thank you again for your kind review and comments. We believe we have addressed your concerns adequately. Since we are still in the discussion process, we therefore respectfully ask you to review our responses once more and determine whether there are any additional concerns. We sincerely hope that we will be able to use the remaining time to engage in an open dialogue with domain experts to enhance the quality of our work.
> > > >
> > > > Thanks for your valuable time in advance.
> > > >
> > > > Authors.

---

> > > > > ### Comment · Reviewer_DXNR · 2023-11-21
> > > > > **Two follow-up questions**
> > > > >
> > > > > - The authors said "addresses this issue from an image-centric standpoint", I do not completely agree because the proposed method can be also adopted for LiDAR modality (e.g., generate sensitivity map of LiDAR BEV map or all points) from my point of view. I just want to if the author solved a unique problem of image-only attacking. If does, what is the problem?
> > > > >
> > > > > - Will the authors open-source the project?
> > > > >
> > > > > I will give an overall evaluation after the authors address the concerns above concisely.

---

> ### Author Response · Authors · 2023-11-21
> **Official Response to the Follow-up Question**
>
> Thanks a lot for your continuous help in improving our paper. We answer your new questions below.
>
> **Q1: If the author solved a unique problem of image-only attacking. If so, what is the problem?**
>
> **A1**: Thank you for your comments and queries. We agree that the proposed method can be adapted for LiDAR modality, which we leave as a future direction. In this context, the unique problem of image-only attack that we solve in this work is to reverse-engineer precise projection functions for varying target object locations and orientations from driving footage, which facilitates faithful scene image synthesis during patch generation (as elucidated in our initial response to Q4).
>
>
> **Q2: Will the authors open-source the project?**
>
> **A2**: Yes, we will open-source the project after the anonymous period. At present, our code with detailed instructions to run are available in the supplementary materials.

---

> > ### Comment · Reviewer_DXNR · 2023-11-21
> > **Questions on proj_x()**
> >
> > I I understand correctly, the so-called "reverse-engineer precise projection functions" should be $proj_x()$. I have two questions about it.
> > - Why the project step ① in Figure 4 is necessary? For example, $d$, $g$, and $\alpha$ are usually known numbers and will not be optimized. If so, why not directly optimize the patch in the 3D world?
> > - $proj_x$ seems to be a basic spatial transformation and projection, so why is it a problem to be solved?
> >
> > Feel free to point it out if I have any misunderstandings.

---

> ### Author Response · Authors · 2023-11-21
> **Official Response to the Follow-up Question**
>
> We are genuinely grateful for your thoughtful considerations, and we address the new questions below:
>
> **Q1: Why the project step ① in Figure 4 is necessary?**
>
> **A1**: We agree that $d$, $g$ and $\alpha$ (the so-called projection parameters) are numbers that will not be optimized, but these parameters are not the same for different scene images in the driving footage, especially for object-oriented attacks. In other words, $proj_x()$ is the projection function for a specific scene image $x$ and the projection parameters (e.g., $d_x$, $g_x$ and $\alpha_x$ ) are also unique for $x$. However, the 2D patch image ($p$) optimized by us should be universally effective across frames, hence we need to use $proj_x()$ to project the 2D patch image $p$ onto each scene image $x$ sampled randomly from the dataset during optimization, simulating the patch's appearance on each scene image $x$ (i.e., $p_x = proj_x(p)$ in Equation 6). Step 1 is necessary to connect the 2D patch image $p$ optimized by us and the randomly sampled scene image $x$ that will be fed to the fusion model, as $p$ should be universally effective across various scene images.
>
> **Q2: why is it a problem to be solved?**
>
> **A2**: Despite the form of $proj_x()$ is a basic spatial transformation and projection, the projection parameters (e.g., $d_x$, $g_x$ and $\alpha_x$) in $proj_x()$ for a specific scene image $x$ is dynamically extracted. We solved the problem of dynamically projecting the patch image $p$ onto the target object in randomly sampled scene images during the patch optimization process. This is accomplished by extracting these unique projection parameters for each scene image from the target object's 3D bounding box that is predicted by the fusion model in benign case.
>
> Please kindly let us know if we have any misunderstanding about your questions. Thanks.

---

> > ### Comment · Reviewer_DXNR · 2023-11-22
> >
> > The authors answer my question clearly. However, I still think "extracting these unique projection parameters" is trivial and can not be regarded as a technical issue.  The authors could make a defense for this.
> >
> > Considering the image-only attacking is reasonable and possesses potential application value, I increased my rating to weak accept.

---

> ### Author Response · Authors · 2023-11-22
> **Thank You for the Feedback**
>
> We express our gratitude for your comments and reevaluation of our contribution. Although extracting those projection parameters may not be technically difficult, we think it is a necessary and novel step for the process of dynamic projection in our proposed optimization approach. Thank you for your valuable feedback and continuous help in making our work better.

---

### Official Review · Reviewer_QA2X · 2023-10-31

**Soundness:** 3 good
**Presentation:** 4 excellent
**Contribution:** 3 good
**Rating:** 6
**Confidence:** 4

**Summary:**

This paper proposes a new approach to attacking fusion models from the camera modality, which successfully compromises advanced camera-LiDAR fusion models and demonstrates the weaknesses of relying solely on fusion for defence against adversarial attacks. The proposed attack framework is based on a novel PointAug method that perturbs the point cloud data and generates realistic-looking adversarial examples. The experiments conducted in both simulated and physical-world environments show the practicality and effectiveness of the proposed approach.

**Strengths:**

1.Sophisticated Camera-LiDAR Fusion Model: The proposed methodology exemplifies a commendable synthesis of advanced camera-LiDAR fusion models. This amalgamation not only taps into the inherent strengths of individual modalities but also crafts a synergistic fusion, ensuring that the combined system is more robust and efficient than its constituent parts in isolation.
2.Efficacy of PointAug in Generating Adversarial Samples: An intrinsic highlight of the paper is the PointAug method, which adeptly fabricates realistic-looking adversarial examples. Such capability is pivotal, particularly in the realm of robust machine learning, as it enables researchers to thoroughly evaluate the resilience of models against potential adversarial threats.
3.Rigorous Experimental Validation in Diverse Environments: The rigorousness and diversity of experiments set this work apart. By conducting evaluations in both simulated and real-world environments, the paper fortifies the assertion of the proposed approach's practicality and efficacy. This dual-pronged validation underscores the method's adaptability and reliability in a wide range of scenarios.

**Weaknesses:**

1.Limited Fusion Model Efficacy: The methodology, while promising, seems to be narrowly tailored for a specific set of fusion models. This raises concerns about its universality. An in-depth exploration into its effectiveness against a broader spectrum of fusion models would have provided a more comprehensive perspective, allowing for a holistic understanding of its potential and pitfalls.
2.Data Dependency and Generalizability Concerns: The experiments, predominantly based on a circumscribed dataset, cast doubts on the model's capacity to generalize across diverse scenarios. The exclusive reliance on a limited dataset can inadvertently introduce biases, thereby undermining the robustness of the approach when deployed in novel, real-world situations.
3.Inadequate Security Analysis and Potential Resource Constraints: While the paper delves into several aspects of the proposed approach, it seems to sidestep a comprehensive analysis of its security implications. Given the pivotal role of security in such contexts, a detailed discourse would have been invaluable. Furthermore, the potentially substantial computational overhead required to generate adversarial samples may render the approach untenable for resource-constrained environments. The paper's omission of a thorough dissection of its inherent limitations further obscures the potential challenges one might encounter in its adoption.

**Questions:**

1.Comparison with Pre-existing Methodologies: Given the emergence and evolution of methodologies targeting fusion models, how does the proposed approach position itself relative to these existing strategies? An analytical juxtaposition against established techniques would elucidate its uniqueness, advantages, and potential shortcomings.
2.Defensive Countermeasures against the Approach: While the paper sheds light on an innovative adversarial approach, it begs the question: What are the viable defensive strategies that can be deployed to counteract its effects? Unveiling potential countermeasures not only underscores the resilience of the approach but also aids in the development of more robust fusion models.
3.Extension and Scalability Concerns: Fusion models are diverse and multifaceted. How malleable is the proposed approach in its application to other fusion model variants? A deeper dive into its adaptability would provide insights into its scalability and flexibility across various fusion paradigms.
4.Implications for Autonomous Vehicle Security: Given the pivotal role of fusion models in autonomous vehicle systems, the proposed adversarial approach inevitably raises safety and security concerns. How might these attacks compromise the integrity and reliability of autonomous driving systems? A comprehensive discussion on this would be crucial for stakeholders in the autonomous vehicle domain.

---

> ### Author Response · Authors · 2023-11-18
> **[Part 1/2] Official Response to Reviewer QA2X (Q1, Q2, Q3 & Q4)**
>
> We express our deepest gratitude for the reviewer’s invaluable time and astute comments. It is our utmost pleasure to address the principal concerns as follows.
>
> **Q1: Limited Fusion Model Efficacy.**
>
> **A1**: Your thoughtful comments are appreciated. As we elaborated in Related Works, Discussion of Other Fusion Strategies (Appendix B) and Limitations (Appendix P), our research primarily focuses on data-level and feature-level fusion strategies. This focus is driven by the fact that advanced fusion models, which demonstrate superior detection performance, predominantly employ these fusion strategies [1]. These strategies are favored due to their enhanced feature extraction capabilities. The adoption of such early-fusion schemes is also a general trend with the increasing popularity of end-to-end autonomous driving [2-3]. Decision-level fusion, the other strategy, is less prevalent in recent fusion models. In such a setup, our camera-only attacks would only influence the output of the camera-based model. Therefore, the impact of such an attack is significantly dependent on the relative weights assigned to the two modalities during the fusion process. This dependency is a common limitation for single-modal attacks. While prior LiDAR-only attacks on Baidu Apollo have shown success [4], this can be attributed to Apollo's current higher weighting of LiDAR results. Enhancing the success rate of camera-only attacks against decision-level fusion models poses an open problem, and we acknowledge it as a promising area for future exploration and research. Thanks for your insightful comments, we have highlighted the fusion strategies we focused on in the introduction of our revised version and discussed more about decision-level fusion in Appendix B.
>
> **Q2: Data Dependency and Generalizability Concerns.**
>
> **A2**: We express gratitude for your valuable feedback. The Nuscenes dataset employed in our experiments is a real-world dataset encompassing diverse scenarios across driving conditions. Models trained on this dataset exhibit robust generalization to real-world and simulation scenarios, as scrutinized in the practicality section of our manuscript. The fusion model, trained with the Nuscenes dataset, successfully detects all objects near the victim vehicle in benign cases. This performance across the Nuscenes dataset, real-world conditions, and simulation environments underscores the generalizability and minimum data dependency of our fusion models. Furthermore, our attacks demonstrate success across three types of data, affirming the generalizability and robustness of our proposed attack approach.
>
> **Q3: Inadequate Security Analysis and Potential Resource Constraints.**
>
> **A3**: We appreciate your insightful comments and suggestions. In response, we have enriched the revised version of our manuscript with a more comprehensive analysis of autonomous vehicle security implications in Appendix R. To address potential resource constraints, we've introduced a detailed threat model for our attack in Appendix Q, outlining the assumptions made about attackers. Additionally, specific devices and the detailed computational overhead required for generating adversarial patches are disclosed in Appendix H, providing a thorough dissection of our approach's resource requirements and inherent limitations.
>
> **Q4: Comparison with Pre-existing Methodologies.**
>
> **A4**: Your comments are valued. In the Related Work section of the original manuscript, we have thoroughly discussed the relationship between our attack and prior attacks against fusion models. Succinctly, some previous attacks are multi-modal attacks that fool camera and LiDAR modalities either separately (with colors and shape) or concurrently (with shape). Others are single-modal attacks against only the LiDAR modality. In contrast, our approach pioneers a single-modal attack exclusively targeting the camera modality. As stated in the Introduction, prior LiDAR-related approaches pose implementation challenges, necessitating additional equipment like photodiodes, laser diodes, or industrial-grade 3D printers to manipulate LiDAR data, thereby increasing deployment costs for attackers. Our camera-only attacks, in comparison, are more affordable and easier to deploy, enabling attackers to print generated adversarial patches with home printers. This reduction in effort heightens the threat to autonomous vehicle security.

---

> > ### Author Response · Authors · 2023-11-18
> > **[Part 2/2] Official Response to Reviewer QA2X (Q5, Q6 & Q7)**
> >
> > **Q5: Defensive Countermeasures against the Approach.**
> >
> > **A5**: We extend our gratitude for your observations. In Appendix O of our original manuscript, we delved into a comprehensive discussion of defensive countermeasures. In summary, architectural-level and DNN-level defense techniques are explored. The architectural-level defense suggests that the unique sensitivity type of different fusion models (i.e., global sensitivity and object sensitivity) results from the choice of image backbone network, and using the CNN-based backbones like ResNet (instead of transformers-based backbones) could minimize the sensitive region to the object-level, making the model robust to scene-oriented attacks. For the DNN-level defense, we have evaluated some directly applicable defensive techniques such as input transformations. Results in Figure 18 indicate that our approach maintains a high attack success rate, even when benign performance is slightly impacted by the defense techniques. Hence, novel adaptations of advanced defenses (like adversarial training) or the development of new defense techniques tailored to sensor fusion models are a direction we propose for future research.
> >
> > **Q6: Extension and Scalability Concerns.**
> >
> > **A6**: Your comments are appreciated. Our approach is not tailored to specific fusion models but rather targets the widely adopted and well-performing fusion strategies of data-level and feature-level fusion. These strategies, utilizing deep neural networks for unified detection results, inherently render them vulnerable to our camera-only single-modal attacks. Appendix D provides a detailed feasibility analysis, illustrating the rationale and foundation of our single-modal attacks. This analysis suggests that our technique is agnostic to specific model designs and easily extensible to various models. The scalability of our technique has also been validated in the evaluations. As stated in Appendix C, our evaluation have covered models with various designs, including data-level (e.g., PointAugmenting) and feature-level (e.g., UVTR, DeepInteraction, BEVFusion-PKU, BEVFusion-MIT, TransFusion) fusion strategies, alignment-based (e.g., UVTR, BEVFusion-PKU, BEVFusion-MIT) and nonalignment-based (e.g., DeepInteraction, Transfusion) fusion techniques for feature-level fusion, and different alignment perspectives like voxel-based (e.g., UVTR) and BEV-based (e.g., BEVFusion-PKU and BEVFusion-MIT) alignments.  Furthermore, our attack extends to camera-based models like BEVFormer in our evaluation.
> >
> > As discussed in Appendix B, the decision-level fusion strategy is less adopted in recent fusion models and not our focus in this work. The impact of our attack on decision-level fusion is significantly dependent on the relative weights assigned to the two modalities during the later fusion process. This dependency is a common limitation for single-modal attacks and we leave the investigation of effective attacks against decision-level fusion strategies as a future direction.
> >
> >
> > **Q7: Implications for Autonomous Vehicle Security.**
> >
> > **A7**: We extend our thanks for your valuable suggestions. Beyond our primary focus on attacking 3D object detection models in this work, we have incorporated a discussion about the implications for autonomous vehicle security in Appendix R of our revised manuscript.
> >
> > **Reference**
> >
> > [1] Motional. Nuscenes Object Detection Leaderboard.
> > https://www.nuscenes.org/object-detection?externalData=all&mapData=all&modalities=Any. 2023.
> >
> > [2] Breakdown: How Tesla will transition from Modular to End-To-End Deep Learning. https://www.thinkautonomous.ai/blog/tesla-end-to-end-deep-learning/  2023.
> >
> > [3] Chen, Li, Penghao Wu, Kashyap Chitta, Bernhard Jaeger, Andreas Geiger, and Hongyang Li. "End-to-end autonomous driving: Challenges and frontiers." arXiv preprint arXiv:2306.16927 (2023).
> >
> > [4] Hallyburton, R. Spencer, Yupei Liu, Yulong Cao, Z. Morley Mao, and Miroslav Pajic. "Security Analysis of {Camera-LiDAR} Fusion Against {Black-Box} Attacks on Autonomous Vehicles." In USENIX Security, 2022.

---

### Official Review · Reviewer_qqbx · 2023-11-01

**Soundness:** 2 fair
**Presentation:** 3 good
**Contribution:** 3 good
**Rating:** 5
**Confidence:** 4

**Summary:**

This paper studies the vulnerability of multi-sensor fusion to adversarial attacks in autonomous driving. The authors propose to leverage the adversarial patch to attack the camera modality in 3D object detection. Specifically, they propose an attack framework employing a two-stage optimization-based strategy that first evaluates vulnerable image areas under adversarial attacks, and then applies dedicated attack strategies for different fusion models to generate deployable patches.

**Strengths:**

- This paper studies an important concern in autonomous driving, i.e., the vulnerability of multi-sensor fusion to adversarial attacks.

- Multiple feature-level fusion models are considered in this paper.

- The performance of the proposed framework is evaluated through both simulated and real-world experiments.

**Weaknesses:**

- This paper does not provide the threat model. What information is available to the attacker during the attack? How feasible is it for the attacker to access this information in a real-world setting? What are the attacker's capabilities?

- The practicality and generalizability of the proposed attack are limited due to its ineffectiveness on decision-level fusion models, which are widely used in many autonomous driving systems, such as Baidu Apollo. Although the proposed attack can alter camera inputs, it fails to affect the outputs of decision-level fusion models. Moreover, these models tend to depend more on LiDAR detection results, further diminishing the practicality of the proposed attack.

- I found it hard to understand the positioning of this paper. There are many existing works studying the attacks against camera-LiDAR fusion models [1,2]. These methods can be used to attack all three types of sensor fusion models including data-level fusion, feature-level fusion, and decision-level fusion. However, the method proposed in this paper can only be used to attack the first two types of fusion models. So, what is the major advantage of this work compared to those existing works? The authors should compare their method with existing attacks to demonstrate superiority of the proposed attack.

[1] Exploring adversarial robustness of multi-sensor perception systems in self driving.
[2] Invisible for both camera and lidar: Security of multi-sensor fusion based perception in autonomous driving under physical-world attacks.

- The practicability of the adopted adversarial patch is questionable. Table 7 shows that the minimum dimensions of the patch are 1 meter by 1 meter, and the patch is too large to be practical. How to place such a large patch on a pedestrian in the real world? In addition, it's impractical to place such a large patch on the back of a vehicle. The patch may hide the license plate of the vehicle, which is prohibited by the traffic law.

- The real-world evaluation is weak. The authors propose to use a patch with a special color pattern to conduct the attack. Such a color pattern can be affected by many factors in the physical world such as light condition, the distance between the camera and the patch, as well as the view angle of the camera. However, the authors do not evaluate the impact of these factors in the real-world setting.

- The impact of the proposed attack on the vehicle's motion remains unclear. Is the perception system of the vehicle consistently deceived by the attack? Is the vehicle's trajectory affected by the attack?

**Questions:**

- What is the threat model. What information is available to the attacker during the attack? How feasible is it for the attacker to access this information in a real-world setting? What are the attacker's capabilities?

- What is the major advantage of this work compared to existing attacks against camera-LiDAR fusion models?

- How to place the adversarial patch on a pedestrian in the real world (as shown in Figure 10)? How to place the patch on the back of a vehicle? Does the patch hide the license plate of the vehicle?

- Does the proposed attack maintain its effectiveness under varying light conditions in the real world?

- The impact of the proposed attack on the vehicle's motion remains unclear. Is the perception system of the vehicle consistently deceived by the attack? Is the vehicle's trajectory affected by the attack?

**Details Of Ethics Concerns:**

The experiment in this paper involves human subjects, but the authors do not report ethical approvals from an appropriate ethical review board.

---

> ### Author Response · Authors · 2023-11-18
> **[Part 1/3] Official Response to Reviewer qqbx (Q1, Q2 & Q3)**
>
> We express our deepest gratitude for the reviewer’s  time and  insightful comments. It is with great respect that we have endeavored to address the primary concerns as follows.
>
> **Q1: What is the threat model of the proposed attack?**
>
> **A1**: We sincerely appreciate your insightful question. Our attack assumes that the attacker has complete knowledge of the camera-LiDAR fusion model used by the target autonomous driving vehicle. Therefore, our attack model is considered to be in a white-box setting. This assumption is congruent with existing endeavors in the literature dedicated to adversarial attacks on autonomous driving systems [1-5]. To achieve this level of knowledge, the attacker may employ methods such as reverse engineering the perception system of victim vehicles [6], employing open-sourced systems, or exploiting information leaks from insiders. In addition, the attacker is clear about the target object and driving scenes he wants to attack. For example, he can record both video and LiDAR data using his own vehicle and devices while following a target object (for object-oriented attacks) or stopping at a target scene (for scene-oriented attacks). Leveraging this pre-recorded data, the attacker undertakes a one-time effort to generate an adversarial patch using our proposed approach. The attacker can then print and deploy the generated patch on the target object or in the target scene for real-time attacks against victim vehicles. The detailed elaboration of our threat model can be found in the appendix Q of our manuscript.
>
> **Q2: The method’s ineffectiveness on decision-level fusion models.**
>
> **A2**: We express our gratitude for your thoughtful comments. As we elaborated in Related Works, Discussion of Other Fusion Strategies (Appendix B) and Limitations (Appendix P) sections, our research primarily focuses on data-level and feature-level fusion strategies. The decision-level fusion is less prevalent in recent fusion models. In such a setup, our camera-only attacks would only influence the output of the camera-based model. Therefore, the impact of such an attack is significantly dependent on the relative weights assigned to the two modalities during the fusion process. This dependency is a common limitation for single-modal attacks. Although prior LiDAR-only attacks have succeeded on Baidu Apollo [10], it can be attributed to the current higher weighting of LiDAR results in Apollo. Our focus on data-level and feature-level fusion is driven by the fact that advanced fusion models, which demonstrate superior detection performance, predominantly employ these fusion strategies [7], due to their enhanced feature extraction capabilities. The adoption of such early-fusion schemes is also a general trend with the increasing popularity of end-to-end autonomous driving [8-9]. The challenge of enhancing the success rate of camera-only attacks against decision-level fusion models remains an open question, and we duly acknowledge this as a promising area for future exploration and research. Based on your comments, we have highlighted the fusion strategies we focused on in the introduction of our revised version and discussed more about decision-level fusion in Appendix B.
>
> **Q3: The positioning of this paper. What is the major advantage of this work compared to existing attacks against camera-LiDAR fusion models?**
>
> **A3**: We are genuinely grateful for your thoughtful considerations. In the Related Work section of the original manuscript, we have discussed the interplay between our work and the works you referenced. In essence, the referenced works encompass multi-modal attacks that deceive  camera and LiDAR modalities either individually (with colors and shape) or concurrently (with shape). In contrast, our focus is on single-modal attacks directed exclusively at the camera modality. As stated in the introduction, prior approaches, as you indicated, present implementation challenges due to additional equipment requirements, such as photodiodes, laser diodes [10], or industrial-grade 3D printers [1], for manipulating LiDAR data. This increases deployment costs for attackers. In contrast, our camera-only attacks offer enhanced affordability and ease of deployment, enabling attackers to print the generated adversarial patch with home printers. This reduction in effort amplifies the threat to autonomous vehicle security.

---

> ### Author Response · Authors · 2023-11-18
> **[Part 2/3] Official Response to Reviewer qqbx (Q4 & Q5)**
>
> **Q4: The practicability of the adopted adversarial patch is questionable. How to place the patch on the back of a vehicle? Does the patch hide the license plate of the vehicle? How to place the adversarial patch on a pedestrian in the real world (as shown in Figure 10)?**
>
> **A4**: We sincerely appreciate your inquiries. Allow us to clarify that the 1m * 1m is not the minimum patch size required for attacks. The minimum patch size necessary for a successful attack is contingent upon the distance between the target object and the victim vehicle. The critical factor is the pixel count influenced by the patch in the input images, and a smaller patch can exhibit efficacy when in close proximity to the victim vehicle.  Table 5 in our study demonstrates that the attack performance of adversarial patches diminishes with increasing distances. Our additional exploration of object-oriented attacks, considering patch size, shape and varying distances between target and victim vehicles, is detailed in Table R1. As shown, the 0.5m * 0.5m patch still performs well at a closer distance (e.g. less than 8m). It is crucial to underscore that failure to detect the target object at a close range is still hazardous, as the driver may lack sufficient reaction time, leading to a braking distance exceeding the actual distance.
>
> Additionally, the patch does not need to be in square shape. Attackers have the flexibility to define patches with rectangular or arbitrary shapes to circumvent covering the target vehicle's license plate.
>
> Figure 10 in Appendix G (Property Validation of Sensitivity Heatmap) aims to demonstrate the property of the sensitivity heatmap instead of real-world attacks against pedestrians. Attackers may use a smaller patch on T-shirts (like [11]) to attack the pedestrians at a closer distance (e.g. 5 m).
>
> The additional experiments and explanations have been added to the revised manuscript in Appendix M.
>
> Table R1: Detection score of the target object under different distances and patch sizes.
>
> | **Distance** |  | 1m$\times$1m | | 1m$\times$0.5m | | 0.5m$\times$0.5m | |
> |:---:|:---:|:---:|:---:|:---:|:---:|:---:|:---:|
> | | **Ben.** | **Adv.** | **Diff.$\uparrow$** | **Adv.** | **Diff.$\uparrow$** | **Adv.** | **Diff.$\uparrow$** |
> | **5m** | 0.711 | 0.074 | **89.59%** | 0.081 | **88.61%** | 0.146 | **79.47%** |
> | **6m** | 0.726 | 0.112 | **84.57%** | 0.108 | **85.12%** | 0.144 | **80.17%** |
> | **8m** | 0.682 | 0.105 | **84.60%** | 0.137 | **79.91%** | 0.191 | **71.99%** |
> | **10m** | 0.643 | 0.127 | **80.25%** | 0.242 | **62.36%** | 0.402 | **37.48%** |
> | **15m** | 0.655 | 0.146 | **77.71%** | 0.48 | **26.72%** | 0.626 | **4.43%** |
>
> **Q5: The real-world evaluation is weak.**
>
> **A5**: We express our gratitude for your observation. In Section 4, our approach accounts for diverse lighting conditions, distances, and viewing angles during the patch optimization process. This is achieved by randomizing brightness, contrast, and projection parameters of the patch and incorporating Estimation of Transformations (EoT) in training.  The evaluation of attack performance under varied distances and viewing angles is meticulously presented in Appendix M. To assess the practicality of our attacks, we conducted experiments in both real-world and high-fidelity simulator environments following the practical attack procedures. Responding to your insightful suggestion, we augmented the practicality evaluation by conducting additional experiments at different times of the day to simulate lighting changes. The experimental settings remain consistent with those detailed in Appendix K, and the results are presented in Table R2.  The second and third columns indicate the benign and adversarial performance of BEVFusion-PKU respectively. The fourth column presents the percentage of model performance degradation, indicating our attack effectiveness. As shown, our patch remains a robust attack performance at different lighting conditions.
>
> Table R2: Attack performance at different times of the day with various lighting conditions.
>
> | Time of the day | Ben. AP | Adv. AP | Difference $\uparrow$ |
> |:---------------:|:---------:|:--------------:|:----------:|
> |     **9:00 AM**     |   0.428   |      0.191     | **55.37%** |
> |     **12:00 PM**    |   0.457   |      0.184     | **59.74%** |
> |     **3:00 PM**     |   0.481   |      0.165     | **65.70%** |
> |     **6:00 PM**     |    0.43   |      0.204     | **52.56%** |

---

> > ### Author Response · Authors · 2023-11-18
> > **[Part 3/3] Official Response to Reviewer qqbx (Q6 & Q7)**
> >
> > **Q6: The impact of the proposed attack on the vehicle's motion remains unclear.**
> >
> > **A6**: We acknowledge your pertinent concern. Indeed, the perception system of the victim vehicle is consistently deceived by our adversarial patch. As demonstrated in the demo video, the adversarial patch will affect the object detection in each data frame, consequently impeding subsequent tasks such as motion tracking and trajectory prediction of target objects within the victim vehicle. This, in turn, elevates the probability of the victim vehicle colliding with undetected targets, thereby posing potentially life-threatening outcomes.  We would like to highlight that we have enriched the revised version of our manuscript with expanded discussions on autonomous vehicle security in Appendix R, extending beyond our primary focus on attacking 3D object detection models.
> >
> >
> > **Q7: Ethics Concerns.**
> >
> > **A7**: We genuinely appreciate your attention to ethical considerations. Our physical-world study involving human subjects underwent thorough scrutiny and approval by an institutional IRB. Notably, we conducted physical experiments in a controlled environment on a closed road, utilizing a camera and tripod to capture scenes instead of employing real cars, as elucidated in Appendix H. This deliberate choice minimizes potential threats to the safety of participants. Stringent protocols were implemented, including participants not facing the camera, wearing masks during experiments, and blurring their faces in the photos. No identifiable information from the volunteers is retained by the researchers. We appreciate your conscientious emphasis on ethical considerations. We have included an ethics statement section (Section 8) to discuss the potential ethical concerns related to the study. We have also sent a message to the Area Chair regarding the ethics concerns.
> >
> > **Reference**
> >
> > [1] Yulong Cao, Ningfei Wang, Chaowei Xiao, Dawei Yang, Jin Fang, Ruigang Yang, Qi Alfred Chen, Mingyan Liu, and Bo Li. Invisible for both camera and lidar: Security of multi-sensor fusion based perception in autonomous driving under physical-world attacks. In IEEE S&P, 2021.
> >
> > [2] Takami Sato, Junjie Shen, Ningfei Wang, Yunhan Jia, Xue Lin, and Qi Alfred Chen. Dirty road can attack: Security of deep learning based automated lane centering under physical-world attack. In USENIX Security 21, 2021.
> >
> > [3] Qingzhao Zhang, Shengtuo Hu, Jiachen Sun, Qi Alfred Chen, and Z Morley Mao. On adversarial robustness of trajectory prediction for autonomous vehicles. In CVPR, 2022
> >
> > [4] Zhu, Zijian, Yichi Zhang, Hai Chen, Yinpeng Dong, Shu Zhao, Wenbo Ding, Jiachen Zhong, and Shibao Zheng. "Understanding the Robustness of 3D Object Detection With Bird's-Eye-View Representations in Autonomous Driving." In CVPR. 2023.
> >
> > [5] Jin, Zizhi, Xiaoyu Ji, Yushi Cheng, Bo Yang, Chen Yan, and Wenyuan Xu. "Pla-lidar: Physical laser attacks against lidar-based 3d object detection in autonomous vehicle." In IEEE S&P, 2023.
> >
> > [6] Fred Lambert, Hacker shows what Tesla Full Self-Driving’s vision depth perception neural net can see, https://electrek.co/2021/07/07/hacker-tesla-full-self-drivings-vision-depth-perception-neural-net-can-see/. 2021.
> >
> > [7] Motional. Nuscenes Object Detection Leaderboard.
> > https://www.nuscenes.org/object-detection?externalData=all&mapData=all&modalities=Any. 2023.
> >
> > [8] Breakdown: How Tesla will transition from Modular to End-To-End Deep Learning. https://www.thinkautonomous.ai/blog/tesla-end-to-end-deep-learning/  2023.
> >
> > [9] Chen, Li, Penghao Wu, Kashyap Chitta, Bernhard Jaeger, Andreas Geiger, and Hongyang Li. "End-to-end autonomous driving: Challenges and frontiers." arXiv preprint arXiv:2306.16927 (2023).
> >
> > [10] Hallyburton, R. Spencer, Yupei Liu, Yulong Cao, Z. Morley Mao, and Miroslav Pajic. "Security Analysis of {Camera-LiDAR} Fusion Against {Black-Box} Attacks on Autonomous Vehicles." In USENIX Security, 2022.
> >
> > [11] Xu, Kaidi, Gaoyuan Zhang, Sijia Liu, Quanfu Fan, Mengshu Sun, Hongge Chen, Pin-Yu Chen, Yanzhi Wang, and Xue Lin. "Adversarial t-shirt! evading person detectors in a physical world." In ECCV, 2020.

---

> > ### Comment · Reviewer_qqbx · 2023-11-21
> > **Follow-up questions**
> >
> > Thanks for the authors' feedback. The following are some follow-up questions.
> >
> > - For an autonomous vehicle equipped with an advanced planning module, would it initiate actions such as changing lanes or braking upon detecting an object at a distance equal to or greater than 8 meters?
> >
> > - I think light conditions in the physical world are more complex. Does the proposed attack maintain its effectiveness under varying lighting conditions in a physical environment?

---

> > > ### Author Response · Authors · 2023-11-22
> > > **Official Response to the Follow-up Questions**
> > >
> > > We express our gratitude to the time and efforts you have dedicated to our rebuttal. We address the new concerns as follows.
> > >
> > > **Q1: Would it initiate actions such as changing lanes or braking upon detecting an object at a distance equal to or greater than 8 meters?**
> > >
> > > **A1**: Your thoughtful comments are appreciated. The conduct of the planning module depends on the speed of the victim vehicle and its relative speed in relation to the target vehicle, as well as surrounding environments. When the two speeds are relatively low, the victim vehicle may not necessarily change lanes or brake in advance before the target vehicle approaches within a certain distance. For instance, in scenarios of traffic congestion, the victim vehicle may follow the target vehicle in front at a close distance (e.g., less than 8 meters) due to their shared low speed. Moreover, in intersections or parking lots, the victim vehicle may halt behind the target vehicle, thus a failure in detection could result in a collision when the victim vehicle starts. We leave the comprehensive examination of advanced planning modules' behavior under attacks to the future work. Your astute comments have been invaluable, and we have incorporated them into our updated discussion on AV security in Appendix R.
> > >
> > > **Q2: Does the proposed attack maintain its effectiveness under varying lighting conditions in a physical environment?**
> > >
> > > **A2**: We acknowledge your pertinent concern. To answer this question, we conducted additional physical-world experiments at different times of the day to study the impact of varying light conditions. We use the same experimental setup as our physical-world experiments detailed in Appendix H. Results are presented in Table R3. As shown, at different times of the day, the average detection scores of the pedestrians are significantly decreased after the patch is deployed, which validates the physical robustness of our patch under different lighting conditions. Thanks for your thoughtful comments, we have added the new experiments in Appendix K of our revised manuscript.
> > >
> > > Table R3: Physical attack performance under various lighting conditions.
> > > | Time | Benign Score | Adversarial Score | Difference $\uparrow$ |
> > > |:---:|:---:|:---:|:---:|
> > > | **9:00 AM** | 0.663 | 0.253 | **61.84%** |
> > > | **6:00 PM** | 0.672 | 0.217 | **67.71%** |

---

### Author Response · Authors · 2023-11-18
**Paper Revision Summary**

We thank all the reviewers for their insightful questions and constructive suggestions! We are glad that the reviewers found our paper “studies an important concern”, “considers sophisticated Camera-LiDAR fusion model”,  “rigorous experimental validation in diverse environments”, “motivation is reasonable”, “extensive experiments with SOTA detectors” and “well written”.

Below is a summary of paper updates, and we also marked the updates in the paper with blue color:

**1. [Section 1: Introduction]** Highlighted our main focus on the early-fusion scheme including data-level and feature-level fusion strategies.

**2. [Section 1: Introduction]** Added reference to the detailed threat model of our attack.

**3. [Section 3: Motivation]** Highlighted our motivation in image-only attacks against fusion models.

**4. [Section 4: Method]** Refined the writing and organization of paragraphs and notations.

**5. [Section 4: Method]** Explained the purpose of the projection function (proj_x()) and the necessity.

**6. [Section 8: Ethics Statement]** Added statements about potential ethical concerns.

**7. [Appendix B: Discussion of Other Fusion Strategies]** Added more discussion on other data-level and decision-level fusion models.

**8. [Appendix H: Detailed Experimental Setups]**  Added descriptions of our training devices and computational overhead.

**9. [Appendix K: Additional Practicality Validation]** Added experiments conducted at different times of the day to validate the impact of various lighting conditions.

**10. [Appendix M: Varying Distances and Viewing Angles]** Added experiments of object-oriented attacks with various patch sizes, shapes and distances from the target object.

**11. [Appendix Q: Threat Model]** Added a detailed threat model for our attacks.

**12. [Appendix R: Discussion on Autonomous Vehicle Security]** Added further discussions on the implications for autonomous vehicle security.

---

### Author Response · Authors · 2023-11-20
**Message from authors**

Dear reviewers,

We appreciate your time and effort in reviewing our work. As we are presently in the discussion phase, we would greatly value the opportunity to engage in further dialogue with you.

Could you kindly let us know if our recent explanations and revisions have addressed your concerns effectively? We hope that these changes have provided a clearer understanding of our contribution. Please rest assured that we are fully prepared to address any additional concerns that may arise.

We eagerly look forward to hearing from you at your earliest convenience.

Thank you once again for your valuable input and guidance.

---

### Meta-Review · Area_Chair_ubd4 · 2023-12-13

**Metareview:**

In this paper, the authors propose a single modal attack framework that could significantly bring down the accuracy of early fusion based AV perception systems (lidar + camera). Specifically, the proposed framework includes a sensitivity recognition stage to categorize the input into either object or global, followed by two different attack strategies. Experiments are conducted on multiple SOTA MSF 3D perception stacks and nuScenes. The results are overall solid and convincing.

A main contribution from this work is the exploration of single modal attack compared to existing ones. The paper does a good job in giving a comprehensive study and revealing many details, upon incorporating the suggestions from the reviewers. The reviews are overall positive, with reviewer qqbx raising many valuable concerns/questions. These concerns and questions are sufficiently addressed by the authors in the rebuttal and revised paper. It is correct as the authors point out that early fusion has become a trend for next generation AV perception, even though many productized AV perception systems are indeed based on late fusion. The observations from the authors are thus interesting and could contribute to the community.

As such, the AC recommends acceptance of the work to ICLR.

**Justification For Why Not Higher Score:**

Despite the contribution, the proposed method is not ground breaking compared to existing ones in terms of the techniques and application scope. The threat model is still focused on white-box attack, which is more limited compared to black-box attack for real-world applications. In addition, as reviewer correctly pointed out, studying the effect of attack on planning behavior output would be even more interesting.

**Justification For Why Not Lower Score:**

This paper is above the ICLR bar for its contribution in method and the solidness in results. The paper does a good job incorporating discussions and details.

---

### Decision · Program_Chairs · 2024-01-16

Accept (poster)